# Homogeneous solution assembled Turing structures with near zero strain semi-coherence interface

Yuanming Zhang[1,2], Ningsi Zhang[1,2], Yong Liu[3], Yong Chen[1,2], Huiting Huang [1,2], Wenjing Wang[1,2], Xiaoming Xu[1,2], Yang Li[1,2], Fengtao Fan[3], Jinhua Ye [4], Zhaosheng Li [1,2✉] & Zhigang Zou [1,2]

Turing structures typically emerge in reaction-diffusion processes far from thermodynamic equilibrium, involving at least two chemicals with different diffusion coefficients (inhibitors and activators) in the classic Turing systems. Constructing a Turing structure in homogeneous solutions is a large challenge because of the similar diffusion coefficients of most small molecule weight species. In this work, we show that Turing structure with near zero strain semi-coherence interfaces is constructed in homogeneous solutions subject to the diffusion kinetics. Experimental results combined with molecular dynamics and numerical simulations confirm the Turing structure in the spinel ferrite films. Furthermore, using the hard-soft acid-base theory, the design of coordination binding can improve the diffusion motion of molecules in homogeneous solutions, increasing the library of Turing structure designs, which provides a greater potential to develop advanced materials.

[1] Collaborative Innovation Center of Advanced Microstructures, National Laboratory of Solid State Microstructures, College of Engineering and Applied Sciences, Nanjing University, 22 Hankou Road, 210093 Nanjing, China. [2] Jiangsu Key Laboratory of Nano Technology, Nanjing University, 22 Hankou Road, 210093 Nanjing, China. [3] State Key Laboratory of Catalysis, Dalian Institute of Chemical Physics, Chinese Academy of Sciences, Dalian National Laboratory for Clean Energy, 116023 Dalian, China. [4] International Center for Materials Nanoarchitectonics (WPI-MANA), National Institute for Materials Science, 1-1 Namiki, Tsukuba 305-0047, Japan. ✉email: zsli@nju.edu.cn

Charge recombination is a major limit in thin films to achieve high-efficiency devices, including inorganic perovskite solar cells, organic solar cells, and photoelectrochemical (PEC) tandem cells, seriously restricting their service life[1–5]. A bulk heterojunction (BHJ) strategy with a high efficiency for charge separation and ultrafast charge transfer increases the power conversion efficiency of solar cells[6–8], which also enables cathodic energy efficiency in $CO_2$ electroreduction[9]. Obviously, selecting a semiconductor to meet some requirements, such as higher carrier mobility, appropriate band structure and interface state, is the key for this kind of BHJ with the perovskite. Similarly, PEC, as a new promising energy technology[10,11], is developing some hierarchical composites to address these issues, such as core-shell heterojunction[12] and layered heterojunction[13]. These approaches have resulted in increased charge separation efficiency, but reduced manufacturing convenience and repeatability due to lattice mismatch, especially for large-area films.

The interface strain increases with the degree of lattice mismatch (Fig. 1a), so interface strain control has been at the forefront of device efficiency enhancement by minimizing undesirable defect formation. Precise and controllable interface assembly is a bridling factor affecting the further development of inorganic perovskite solar cells, organic solar cells and other fields. Numerous natural systems contain surfaces or threads that enable directional interface contact[14,15], such as zebra stripes, spot fish, and undulating sand dunes (Fig. 1b). Some spatiotemporal stationary structures, which are summarized by Alan Turing as the interaction of reaction and diffusion with each other[16]. Turing

explained the emergence of stationary patterns by invoking the interplay between an activator and an inhibitor with different diffusion rates[17], which has proven to be extremely influential across many disciplines[18–22] because they can show enhanced features resulting from abundant and smooth interfaces[23]. Recently, Tan[24] used a facile route based on aqueous-organic interfacial polymerization to generate Turing-type polyamide membranes for water purification, and these membranes exhibit excellent water-salt separation performance. Zhang[22] reported a cation exchange approach in the heterogeneous solvent of diethylenetriamine and deionized water to produce Turing-type $Ag_2Se$ on $CoSe_2$ nanobelts relied on diffusion-driven instability, which was highly effective in catalyzing the oxygen evolution reaction (OER) in alkaline electrolytes with an 84.5% anodic energy efficiency. It should be noted that the above case studies the related potential of the Turing structure in heterogeneous solution. However, to date, realization of an inorganic Turing structure (spots or stripes) in homogeneous solutions has not been reported because of the similar diffusion coefficients for most small molecule weight species[24].

Here, we show that the Turing structure with near zero strain semi-coherence interfaces are constructed in homogeneous solutions subject to the diffusion kinetics. Our experimental results combined with molecular dynamics (MD) and numerical simulations confirm the Turing structure in the spinel ferrite films. By using a Zn-Fe-O film as a model, the Turing structure film yields a 49-fold and 5-fold enhancement in charge separation efficiency at 1.6 V versus (vs) the reversible

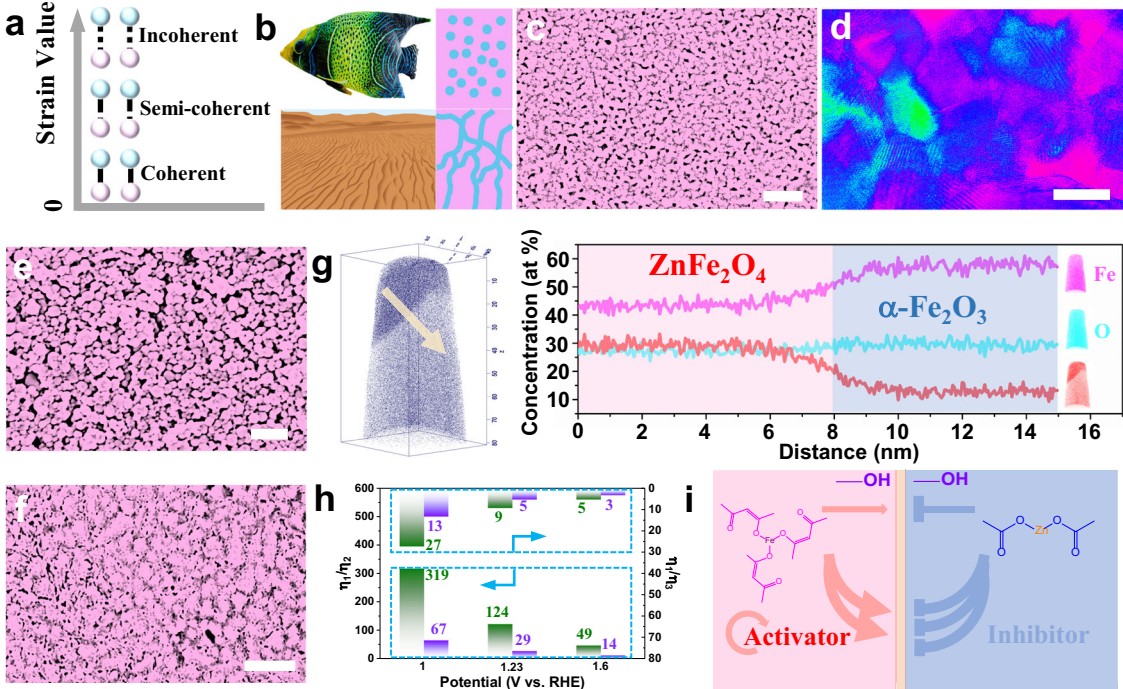

**Fig. 1 Characterization of the Turing structures. a** Interface strain due to lattice mismatch. **b** Turing structures. **c** Scanning electron microscopy (SEM) image of the Turing structure for the Turing interface film (Zn: Fe=1:3) employing iron (III) 2,4-pentanedionate ($Fe[C_5H_7O_2]_3$). **d** False-colored high-angle annular dark-field scanning transmission electron microscopy (HAADF-STEM) image of the Turing interface film in **c**. Purple represents α-$Fe_2O_3$, and green represents $ZnFe_2O_4$ by crystal plane analysis. **e** SEM image of the conventional dual-phase interface film (Zn: Fe = 1:3) (dual phases coexist, but there is no reaction diffusion at the interface). **f** SEM image of the non-Turing dual-phase interface film (Zn: Fe = 1:3) employing ferric nitrate ($Fe(NO_3)_3$) (dual phases coexist, but the gap in reaction diffusion kinetics is small). **g** Three-dimensional APT volume (20 × 20 × 80 $nm^3$) of the center of the Turing interface film (Zn: Fe = 1:3) and one-dimensional concentration profile taken along the arrow. **h** Ratio of the separation efficiencies of the Turing interface film ($\eta_1$) and the conventional dual-phase interface film ($\eta_2$) and non-Turing dual-phase interface film ($\eta_3$). **i** Assuming that the kinetic pathways depend on competing activation ($Fe[C_5H_7O_2]_3$) (slow diffusion of an activator, coordination state) and inhibition ($Zn^{2+}$) (fast diffusion of an inhibitor, ionic state). Scale bars: **c** 100 nm; **d** 20 nm; **e** 100 nm; **f** 200 nm.

hydrogen electrode (RHE) via an enhanced interface built-in electric field compared to that of the conventional dual-phase interface and non-Turing dual-phase interface, respectively.

## Results

**Sample characterization**. For spinel ferrite films, the performance in light-driven water oxidation has remained poor compared with traditional semiconductors in past progress (Supplementary Figs. 1–3). Here, we described that Turing structure with near zero strain semi-coherence interfaces were constructed in homogeneous solutions subject to the diffusion kinetics, which was named the Turing interface at the atomic level. A typical staggered spot microstructure between the dual phases (Fig. 1c, d) was exhibited. In contrast, directionally manufactured conventional dual-phase interface films were also explored in detail through a variety of approaches and displayed a nonuniform microstructure (Fig. 1e and Supplementary Figs. 4 and 5). As shown in Fig. 1f and Supplementary Fig. 6a–f, we did not detect an obvious Turing pattern in the non-Turing dual-phase interface film even though dual phases existed (Supplementary Fig. 6g). The distribution of the dual phases was messy and embedded, which led to the general performance reported for that material (Supplementary Fig. 6h). Furthermore, three-dimensional atom probe tomography (APT) was conducted to investigate the local chemical composition and accurate interface of the Turing interface film. First, aluminum was coated on the surface of the film before making the APT tip samples to prevent damage to the sample during testing, as shown in Supplementary Fig. 7. Then, a significant interface including $ZnFe_2O_4$ and $\alpha$-$Fe_2O_3$ phases in three dimensions was observed (Fig. 1g), which existed widely in bulk (Supplementary Fig. 8). Moreover, there was no parasitic phase in the $ZnFe_2O_4$ film (see detailed information in Supplementary Fig. 9), which indicated that the Turing structure was not caused by traditional phase separation. A remarkable improvement was observed, quantified from separation efficiency-bias voltage curves (Fig. 1h and Supplementary Fig. 10), in the Turing interface film relative to the conventional dual-phase interface film and non-Turing dual-phase interface film. The charge separation efficiency was increased by approximately two orders of magnitude. According to the theoretical basis of the Turing structure[24], we assume that the kinetic pathways depend on competing activation ($Fe[C_5H_7O_2]_3$) (slow diffusion of an activator, coordination state) and inhibition ($Zn^{2+}$) (fast diffusion of an inhibitor, ionic state), which refers to the theoretical model of reaction diffusion (Fig. 1i).

We used Rietveld refinement of X-ray diffraction (XRD) data to characterize the phase composition and ratio (Supplementary Fig. 11), which revealed that the Turing interface film (Zn: Fe = 1:3) contained the mixed phases of $ZnFe_2O_4$ (JCPDS No. 22-1012) and $\alpha$-$Fe_2O_3$ (JCPDS No. 33-0664). The content of $\alpha$-$Fe_2O_3$ is ~31% (R = 9.78%) (Supplementary Fig. 11b), which is very close to the proportion added, indicating the homogeneity and stability of the preparation process. Similarly, the content of $\alpha$-$Fe_2O_3$ is ~57% (R = 8.23%) for the Turing interface film (Zn: Fe = 1:4) (Supplementary Fig. 11c). X-ray photoelectron spectroscopy (XPS) was further employed to study the surface chemical compositions of the samples (Supplementary Figs. 12 and 13), which are consistent with the proportion of the original added materials. The above results reflected the homogeneity of the solution employed; thus, the film morphology was well controlled.

**A re-coordinate mechanism and diffusion coefficient**. In our initial study, we found that the organic iron source and inorganic iron source showed different solvation characteristics (Supplementary Fig. 14). The coordination bond in $Fe[C_5H_7O_2]_3$ can be

opened by other inorganic ions, which is attributed to the chemical hardness that has been revealed with detailed experimental and theoretical calculations (Fig. 2). We calculated the energy levels of different molecular orbitals inspired by the hard-soft acid-base theory[25]. To understand the structure of molecules, we have calculated the molecular electrostatic potential surface (Fig. 2a). The acceptor atoms with negative potential in the red regions represent the most electronegative potential (electrophilic sites), and they are mainly localized over anionic groups. Correspondingly, the electrophilic sites of $[C_5H_7O_2]^-$ and $CH_3COO^-$ are both located around the oxygen atoms and present good confirmation of the close-to-positive charge. Furthermore, the chemical hardness of each ion and ion group are given ($Cl^- > CH_3COO^- > NO_3^- > [C_5H_7O_2]^-$, $Fe^{3+} > Zn^{2+}$) according to theoretical calculations and the Koopman approximation[26] ($\eta = (\epsilon_{LUMO} - \epsilon_{HOMO})/2$, $\eta = (I - A)/2$, where I and A are the ionization energy and electron affinity, respectively) (Fig. 2b, c).

In short, $CH_3COO^-$ will combine with $Fe^{3+}$ ions, and $[C_5H_7O_2]^-$ will re-coordinate completely with $Zn^{2+}$ in a mixed solution of $Fe[C_5H_7O_2]_3$ and zinc acetate ($Zn(CH_3COO)_2$) (Fig. 2d). However, there was no recombination in the mixed solution of ferric chloride ($FeCl_3$) and $Zn(CH_3COO)_2$. In contrast, a small part of the mixed solution of $Fe(NO_3)_3$ and $Zn(CH_3COO)_2$ recombined (Fig. 2e). The Tyndall effect proves that three anions ($Cl^-$, $CH_3COO^-$, $NO_3^-$) can break the complexation of $Fe[C_5H_7O_2]_3$ (Supplementary Fig. 14). More importantly, $[C_5H_7O_2]^-$ will reduce the diffusion rate of $Zn^{2+}$ since $Zn[C_5H_7O_2]_2$ is present in a coordinated form, while more $Fe^{3+}$ ions are released from the $Fe[C_5H_7O_2]_3$, accelerating the diffusion. Furthermore, the diffusion coefficients show orders of magnitude differences (Fig. 2f). Therefore, through chemical re-coordination, the system reached the appropriate difference in the diffusion coefficients of the activator and the inhibitor, resulting in instability of the diffusion drive and the generation of spots and stripes of the Turing structures (Fig. 2g). Such iron complexes, including ferrocene ($Fe[C_5H_5]_2$) and iron glycinate ($Fe[C_2H_4O_2N]_2$), may also possess such characteristics.

**Theoretical simulation of Turing structure formation**. Alan Turing mathematically showed that the stable state can destabilize under certain conditions and spontaneously generate spatially stationary patterns in a reaction-diffusion system, which is particularly well known in mathematical biology. Thus, the Turing pattern can be described by dimensionless reaction-diffusion equations combined with our chemical system:

$$\begin{cases} \frac{\partial u}{\partial t} = f(u,\ v) + d_u \triangle u \\ \frac{\partial v}{\partial t} = g(u,\ v) + d_v \triangle v \end{cases} \quad (1)$$

where $u$ and $v$ is the vector of concentration, the functions $f(u,\ v)$ and $g(u,\ v)$ represent the reaction kinetics (diffusion term), and $d_u$ and $d_v$ are the diffusion coefficients of $u$ and $v$, respectively. In these systems, there are two chemical reactants that can not only interact, but also diffuse alone. Hence, the generation of the Turing pattern corresponds to the coupling of a nonlinear reaction kinetic process and a special diffusion process, which will be unstable due to the different diffusion velocities of the two factors. To demonstrate the instability caused by this diffusion, we continue to analyze this difference between the diffusion coefficients of the two substances by means of a combination of experiments and theory. As shown in Fig. 3a, it should be noted that the diffusion coefficient of $(CH_3COO)^-$ was 4 times higher than the diffusion coefficient of $[C_5H_7O_2]^-$ in deuterated methanol for the 2D DOSY test due to this difference in molecular weight.

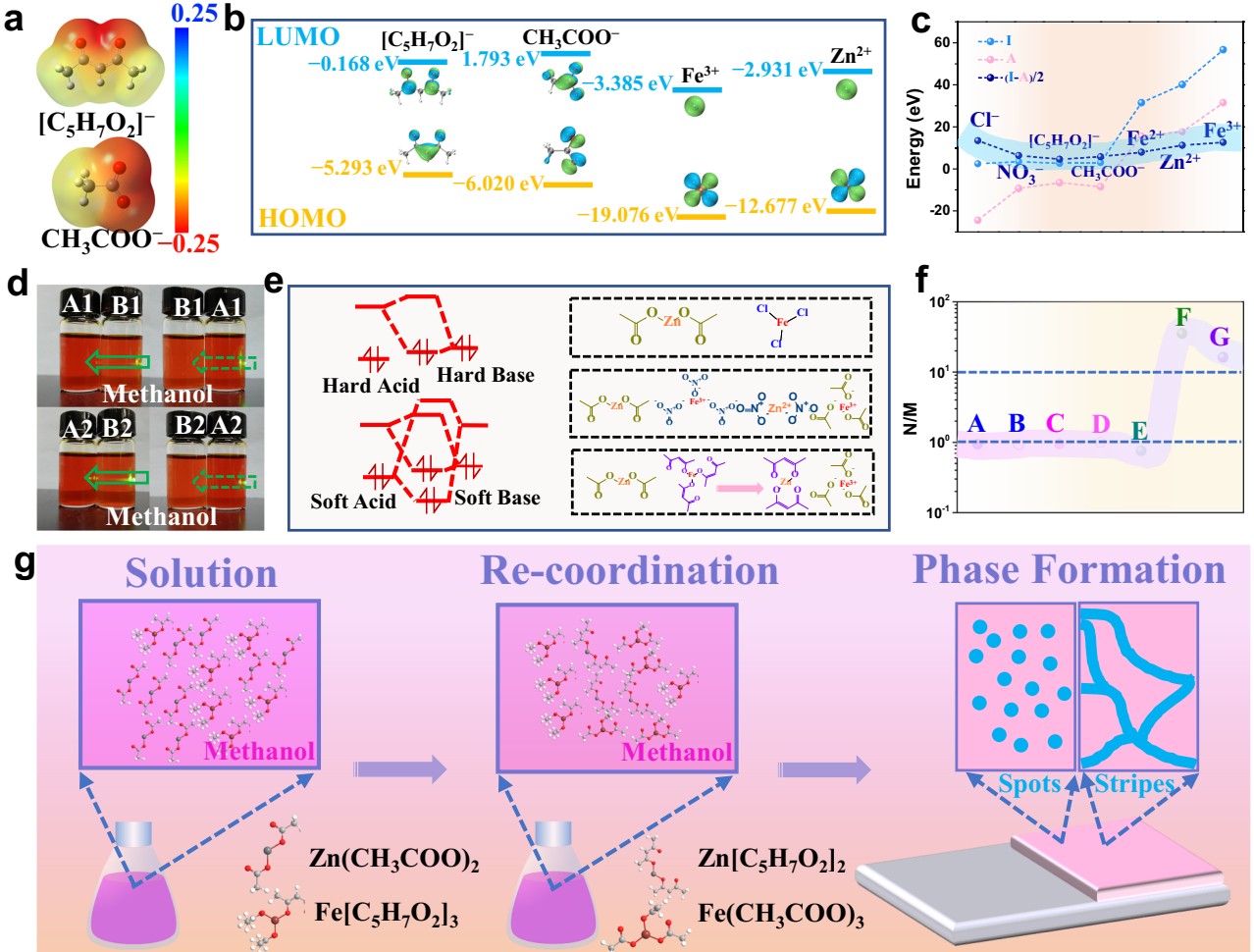

**Fig. 2 Hard-Soft-Acid-Base (HSAB) theory. a** Molecular electrostatic potential surface of $CH_3COO^-$ and $[C_5H_7O_2]^-$. **b** Frontier molecular orbitals for $[C_5H_7O_2]^-$, $CH_3COO^-$, $Fe^{3+}$ and $Zn^{2+}$ obtained by the B3LYP/6-311 G (d, p) method. **c** The ionization energy and electron affinity. **d** The solution exhibits an obvious Tyndall effect upon the addition of the same concentration of acetate ions (A1: 10 mg $Fe[C_5H_7O_2]_3$, 4.5 mL methanol; B1: 7 mg $Ba(CH_3COO)_2$, 10 mg $Fe[C_5H_7O_2]_3$, 4.5 mL $CH_3OH$; A2: 10 mg $Fe[C_5H_7O_2]_3$, 4.5 mL methanol; B2: 5.2 mg $CH_3COOK$, 10 mg $Fe[C_5H_7O_2]_3$, 4.5 mL methanol). **e** Ion exchange reaction based on the HSAB theory. **f** Ratio of diffusion coefficients (Supplementary Fig. 15, for $FeCl_3$, A is Zn: Fe = 1:3 and B is Zn: Fe = 1:4; for $Fe(NO_3)_3$, C is Zn: Fe = 1:3 and D is Zn: Fe = 1:4; for $Fe[C_5H_7O_2]_3$, E is Zn: Fe = 1: 2, F is Zn: Fe = 1:3, and G is Zn: Fe = 1:4). **g** Schematic diagram of Turing structure formation (spots or stripes).

To gain an atomic-level understanding of the re-coordination mechanism in homogeneous solutions, theoretical investigations, including MD simulations, were explored. First, MD simulations are performed to study the dynamic process of $Zn^{2+}$ in various solutions, and there is an order of magnitude difference in the diffusion coefficients of $Zn^{2+}$ from $Zn(CH_3COO)_2$ in the $Fe(NO_3)_3$ and $Fe[C_5H_7O_2]_3$ solutions (Fig. 3b). Meanwhile, it is clear that $Zn^{2+}$ and $[C_5H_7O_2]^-$ show much higher affinity than $CH_3COO^-$ (Fig. 3c), meaning that the dynamic process of $Zn^{2+}$ is slow. While the movement of $Zn^{2+}$ and $Fe^{3+}$ was not affected by $NO_3^-$ (Fig. 3d). The above experiments and MD results provided solid evidence that high molecular weight organic anions play a key role in homogeneous solution for formation of Turing patterns. Accordingly, we confirmed that the kinetic pathways depend on competing activation ($Zn[C_5H_7O_2]_2$) (slow diffusion of an activator, coordination state), and inhibition ($Fe^{3+}$) (fast diffusion of an inhibitor, ionic state) refers to the theoretical model of reaction diffusion.

In our experiment, as the concentration of $Fe[C_5H_7O_2]_3$ increased, the $\alpha$-$Fe_2O_3$ phase became more stripe-like, and $ZnFe_2O_4$ phase existed as spinel octahedral nanoparticles (Fig. 3e–g and Supplementary Figs. 16–18). However, one cannot

claim the Turing pattern just because it looks like the Turing pattern, and the formation of the Turing structure follows the reaction-diffusion equation mentioned earlier. Researchers have found examples of Turing-like mechanisms in the distribution of species in ecosystems, such as the predator-prey model. In our experiment, substance A (Fe) and substance B (Zn) with large difference in diffusion coefficient correspond to inhibitor and activator, respectively. The two substances are simultaneously affected by cross-diffusion according to the hard-soft acid-base theory. At the same time, the reacted substance concentrations are all positive values. These features fit best with classical predator-prey models. More importantly, when conducting relevant research in the fields of economy, physics and chemistry, scholars often select predator-prey model and convert the differential equation into difference equation for numerical simulation[27,28]. The difference equation can well continue some properties of the differential equation, such as instability. Although other models can also be numerically simulated, they are difficult to match the complex conditions of chemical reactions. The prey acts as an activator, seeking to reproduce and increase their numbers, while the predator acts as an inhibitor, keeping populations in check[27]. Since the mathematical

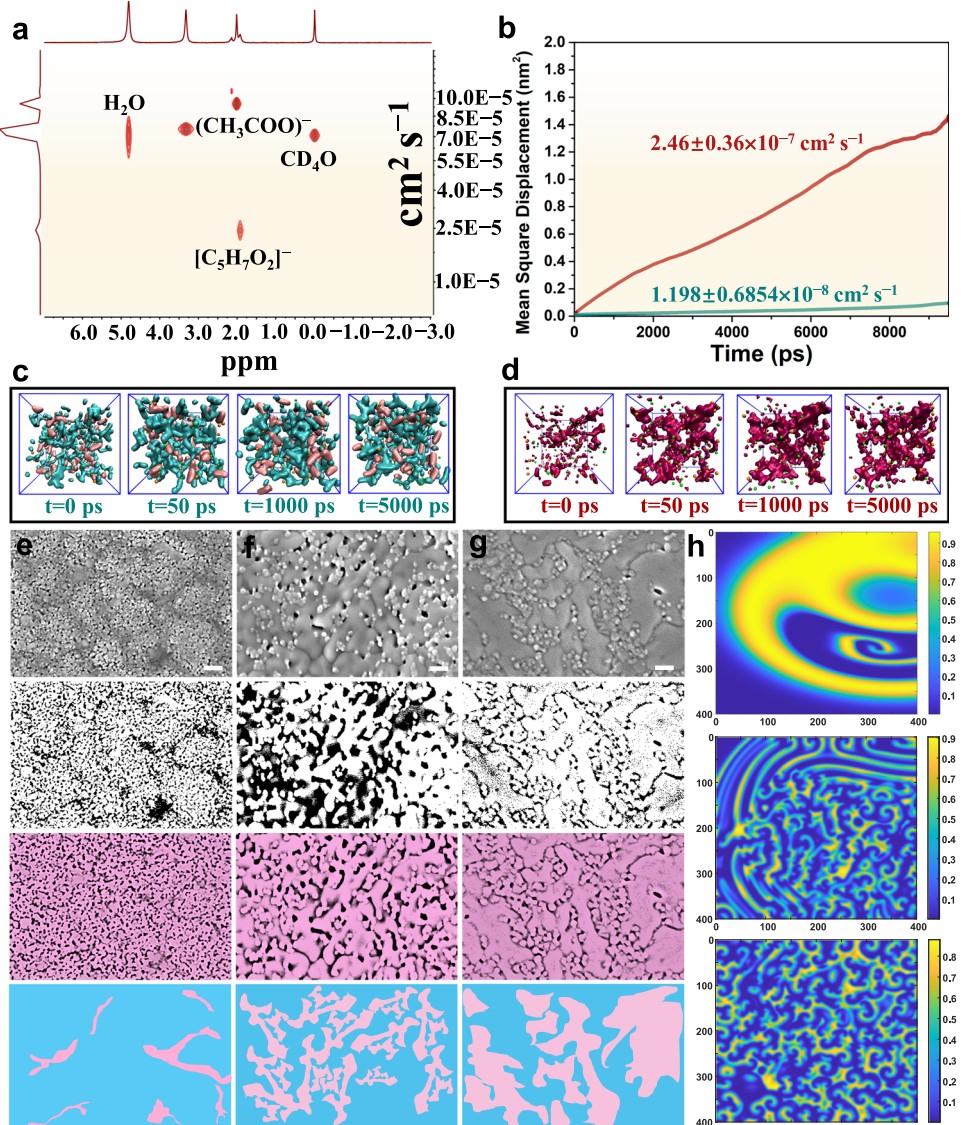

**Fig. 3 Understanding the Turing structure by the reaction-diffusion equation. a** Two-dimensional diffusion-ordered nuclear magnetic resonance spectroscopy (2D DOSY) of a mixed solution of $Fe[C_5H_7O_2]_3$ and $Zn(CH_3COO)_2$. **b** Mean square displacement of $Zn^{2+}$ in different solutions in Fig. 2f. The purple red and dark green curves represent the C solution ($Fe(NO_3)_3$, $Zn(CH_3COO)_2$) and F solution ($Fe[C_5H_7O_2]_3$, $Zn(CH_3COO)_2$), respectively. **c** Snapshots of the dynamic process of F solution ($Fe[C_5H_7O_2]_3$, $Zn(CH_3COO)_2$) at different simulation times. The orange balls are $Zn^{2+}$, the green balls are $Fe^{3+}$, the pink represents $[C_5H_7O_2]^-$ and the cyan represents $CH_3COO^-$. **d** Snapshots of the dynamic process of C solution ($Fe(NO_3)_3$, $Zn(CH_3COO)_2$) at different simulation times. The orange balls are $Zn^{2+}$, the green balls are $Fe^{3+}$, and purple red represents $NO_3^-$. **e** SEM image of Zn: Fe = 1:2.4 film. **f** The Turing interface film (Zn: Fe = 1:3.5). **g** The Turing interface film (Zn: Fe = 1:4). In the fourth row of the figure, blue represents $ZnFe_2O_4$, and pink represents $\alpha$-$Fe_2O_3$. **h** Patterns exhibited by prey and predator of the system at $T = 100$, 600, and 1000, respectively. The difference in diffusion coefficient is 30 times. Scale bars: **c**, **d**, **e**, 100 nm.

equations involved in the above model are coupled nonlinear reaction-diffusion equations, it is difficult to obtain their exact solutions. Therefore, we use the finite difference method to carry out the Turing pattern simulations through MATLAB[28] (detailed formula derivation and parameter setting are shown in the Supplementary Note 1. Numerical Simulation in Supplementary Information). We are now in a position to examine patterns generated from numerical simulations of the system as long as the system parameters change in the Turing space. Thus, for a given $d = \frac{d_B}{d_A} = \frac{d_{Zn}}{d_{Fe}} = \frac{1}{30}$, we expect to see the formation of patterns. Furthermore, the given parameter value according to our chemical system will fall into the Turing space, giving rise to different spatial patterns with respect to time, as shown in Fig. 3h. Turing patterns can still be obtained by changing key parameters

such as diffusion coefficient difference ($d$) and initial conditions (Supplementary Fig. 19). The consistency between experiments and simulations verifies that the formation of the architecture is closely related to Turing's theory. A movie for the numerical simulations of the Turing structures is shown in Supplementary Movie 1.

**In situ optical observation of the shape evolution of the droplets.** The nucleation and crystal growth of spinel ferrite in solution are largely uncontrollable. The larger the area is, the harder it will be to achieve a uniform crystalline film. However, the crystalline spinel ferrite film exhibits a uniform Turing pattern, which is probably related to the evolution of the droplet shape in the early stage. Thus, experiments were carried out under batch conditions. We used optical microscopy to monitor

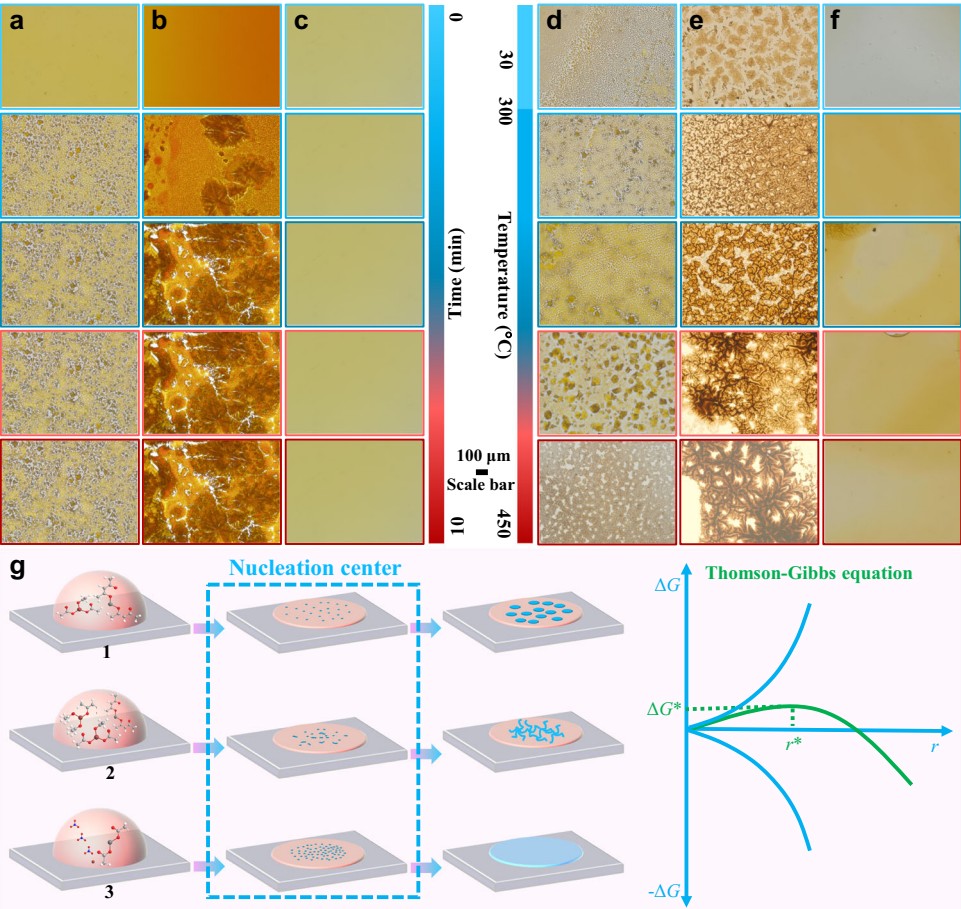

**Fig. 4 In situ optical observation of shape evolution of the droplets. a–c** Shape evolution of the droplets on the FTO for for solutions F, G, and C (Fig. 2f) at room temperature, respectively. **d–f** Shape evolution of the droplets on FTO for solutions F, G, and C at different temperatures. **g** Schematic diagram of structure formation caused by the difference in diffusion coefficient (**1** means large, **2** means medium, and **3** means small).

droplet movement along the fluorine-doped tin oxide (FTO) coated glass surface (Fig. 4). It can be inferred that a spot-like Turing structure is formed due to the difference in the diffusion kinetics of the activator and the inhibitor with the gradual volatilization of the methanol solvent during the natural drying process (Fig. 4a). When the difference in diffusion coefficient decreases, a stripe-like Turing structure is formed, possibly due to the fusion of the spot boundary (Fig. 4b). In contrast, smooth films without any borders were observed employing $Fe(NO_3)_3$ (Fig. 4c) and $FeCl_3$ (Supplementary Fig. 20a). A movie of the shape evolution of droplets is provided as a newly added Supplementary Movie 2.

The speed of solvent volatilization will increase with temperature, but there is no specific shape evolution since spots (Fig. 4d) and stripes (Fig. 4e) still exist. In the cases of $Fe(NO_3)_3$ (Fig. 4f) and $FeCl_3$ (Supplementary Fig. 20b), the temperature rise will cause thermal rupture of the films, instead of forming a Turing structure. Considering the above process comprehensively, we can infer that when difference in the diffusion kinetics of the activator and the inhibitor are very large, the droplets will form multiple nucleation centers (Fig. 4g, process 1), and after the difference is reduced, the nucleation centers will be connected into stripes (process 2). When the difference is very small, uniform nucleation makes the film a smooth and flat structure without the Turing patterns (process 3). The shape evolution of droplets on other substrates, including indium tin oxide (ITO)-coated glass and ordinary glass (Supplementary Fig. 21), was almost consistent with that on FTO. We inferred that the droplet

behavior was consistent with the microscopic nucleation behavior. Therefore, the new droplet smaller than the critical size tended to shrink until it disappeared according to the Thomson-Gibbs equation.

**Interfacial strain analysis**. A well-defined structure-property relationship allows us to better design catalysts. In the $ZnFe_2O_4$ spinel framework, $M_T$-O (tetrahedral) and $M_O$-O (octahedral) are a result of the orbital overlap between the metal $d$ orbitals and oxygen $p$ orbitals (Fig. 5a). Each oxygen is shared by both tetrahedral and octahedral cations, resulting in $M_T$-O or $M_O$-O being weaker[29]. Therefore, it is convenient to graft other types of cations onto oxygen ions. A series of films were prepared using a spray pyrolysis system (Supplementary Fig. 22). Focused ion beam was used to prepare samples for HAADF-STEM (Supplementary Fig. 23), which revealed an atomic-level interface between $ZnFe_2O_4$ (111) and $\alpha$-$Fe_2O_3$ (001), marked with a purple dotted line (Supplementary Fig. 24a, g). We did not detect other crystal planes in the Turing interface film by selected-area electron diffraction (SAED) patterns (Supplementary Fig. 24b–d, h–j). Abundant and smooth atomic interfaces were woven into the Turing structures.

Geometric phase analysis (GPA) based on the HAADF-STEM image of the Turing structure (Fig. 5b) showed that the strain values ($\varepsilon_{xx}$, $\varepsilon_{xy}$, $\varepsilon_{yy}$) were approximately zero at the interface, which were much smaller than those of the conventional dual-phase interface (Fig. 5c). Powder was also prepared following the same process (Supplementary Figs. 25 and 26) in order to further

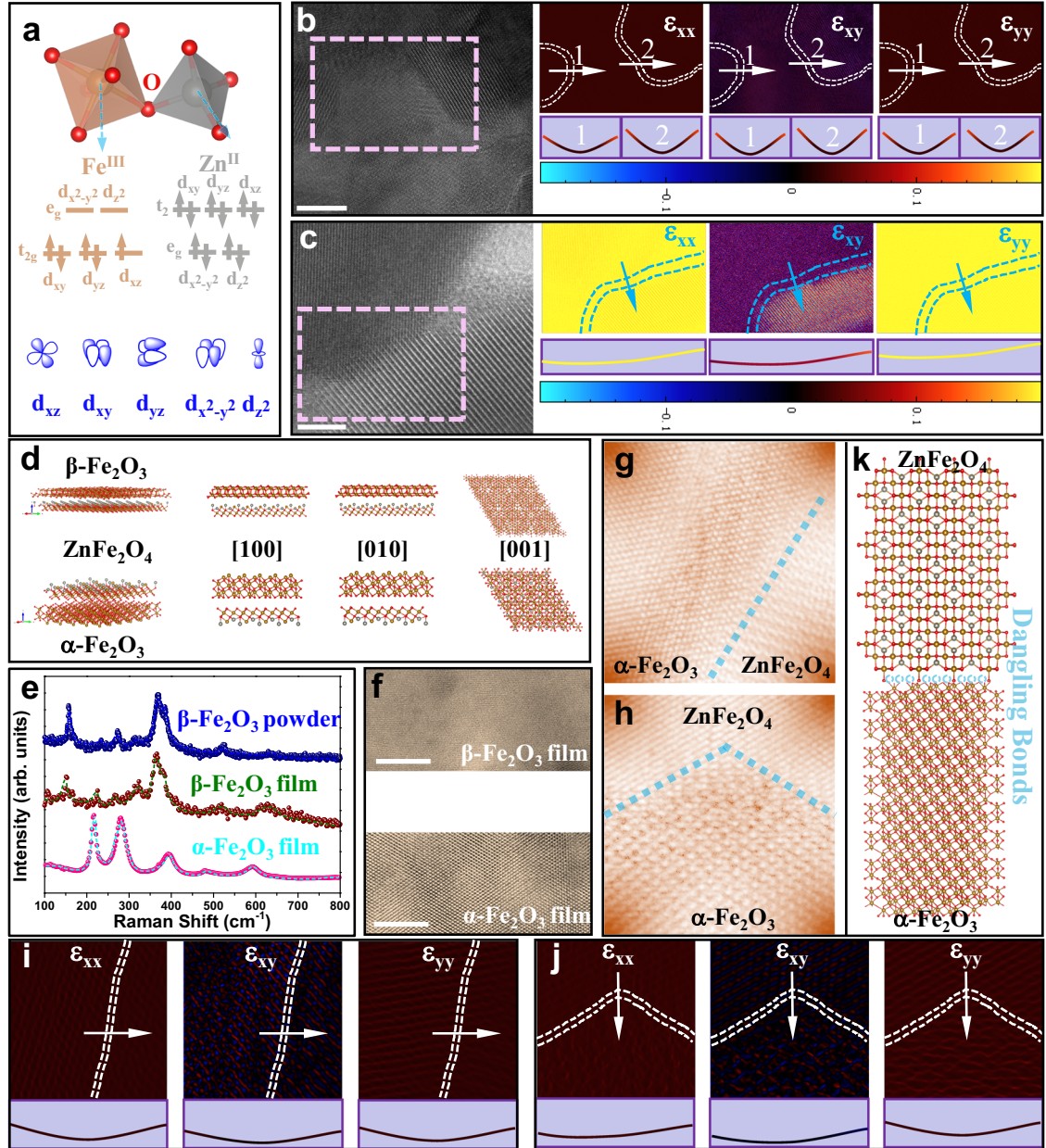

**Fig. 5 Atom-level Turing interface. a** The oxygen-bridged coordination network. **b** HAADF-STEM image of the Turing interface (Zn: Fe = 1:3) and GPA analysis of different plane strains. The interface is marked with a white dashed line in the strain graph, and the curve directly below the graph corresponds to the strain value across the interface. **c** TEM image of the conventional dual-phase interface (Zn: Fe = 1:3) and GPA analysis of different plane strains. The interface is marked with a purple dashed line in the strain graph, and the curve directly below the graph corresponds to the strain value across the interface. **d** Crystallographic models of the interface before the phase transition and after the phase transition. **e** Raman spectra for the β-Fe$_2$O$_3$ powder, α-Fe$_2$O$_3$ powder and the unannealed film (β-Fe$_2$O$_3$). **f** HAADF-STEM images for the unannealed and calcined films for the only iron source (Fe[C$_5$H$_7$O$_2$]$_3$). **g** Image of a larger view in the dotted box (Supplementary Fig. 28a). **h**, Image of a larger view in the dotted box (Supplementary Fig. 28b). **i** GPA analysis of different plane strains for **g**. The interface is marked with a white dashed line in the strain graph, and the curve directly below the graph corresponds to the strain value across the interface. **j** GPA analysis of different plane strains for **h**. The interface has been marked with a white dashed line in the strain graph, and the curve directly below the graph corresponds to the strain value across the interface. **k** The semi-coherent interface. Scale bars: **b**, 5 nm; **c**, 10 nm; **f**, 5 nm.

explore the formation of the interface. Metastable cubic β-Fe$_2$O$_3$ (Fig. 5d, e and Supplementary Fig. 27a) was formed on the FTO during the pyrolysis of Fe[C$_5$H$_7$O$_2$]$_3$ and then transformed into stable hexagonal α-Fe$_2$O$_3$ during the heat treatment.

ZnFe$_2$O$_4$ and β-Fe$_2$O$_3$ have similar crystal structures and the same Fe-O bond lengths. Thus, the lattice mismatch of the dual-phase interface is quite low (δ = 10.8%). A thermally induced phase transition from β-Fe$_2$O$_3$ to α-Fe$_2$O$_3$ occurred, which maintained a low lattice mismatch (δ = 16.9%) as the semi-

coherent interface (Supplementary Fig. 27b). The HAADF-STEM image (Fig. 5f) of the film prepared by spraying using Fe[C$_5$H$_7$O$_2$]$_3$ showed the β-Fe$_2$O$_3$ were included before annealing, and stable α-Fe$_2$O$_3$ was formed after annealing. Abundant α-Fe$_2$O$_3$ (001)/ZnFe$_2$O$_4$ (111) interfaces were confirmed (Fig. 5g, h and Supplementary Fig. 28). The crystallographic parameters of the relevant phases are shown (Supplementary Fig. 29). GPA based on the HAADF-STEM image of the Turing structure (Fig. 5i, j) showes that the strain values (ε$_{xx}$, ε$_{xy}$, ε$_{yy}$) are

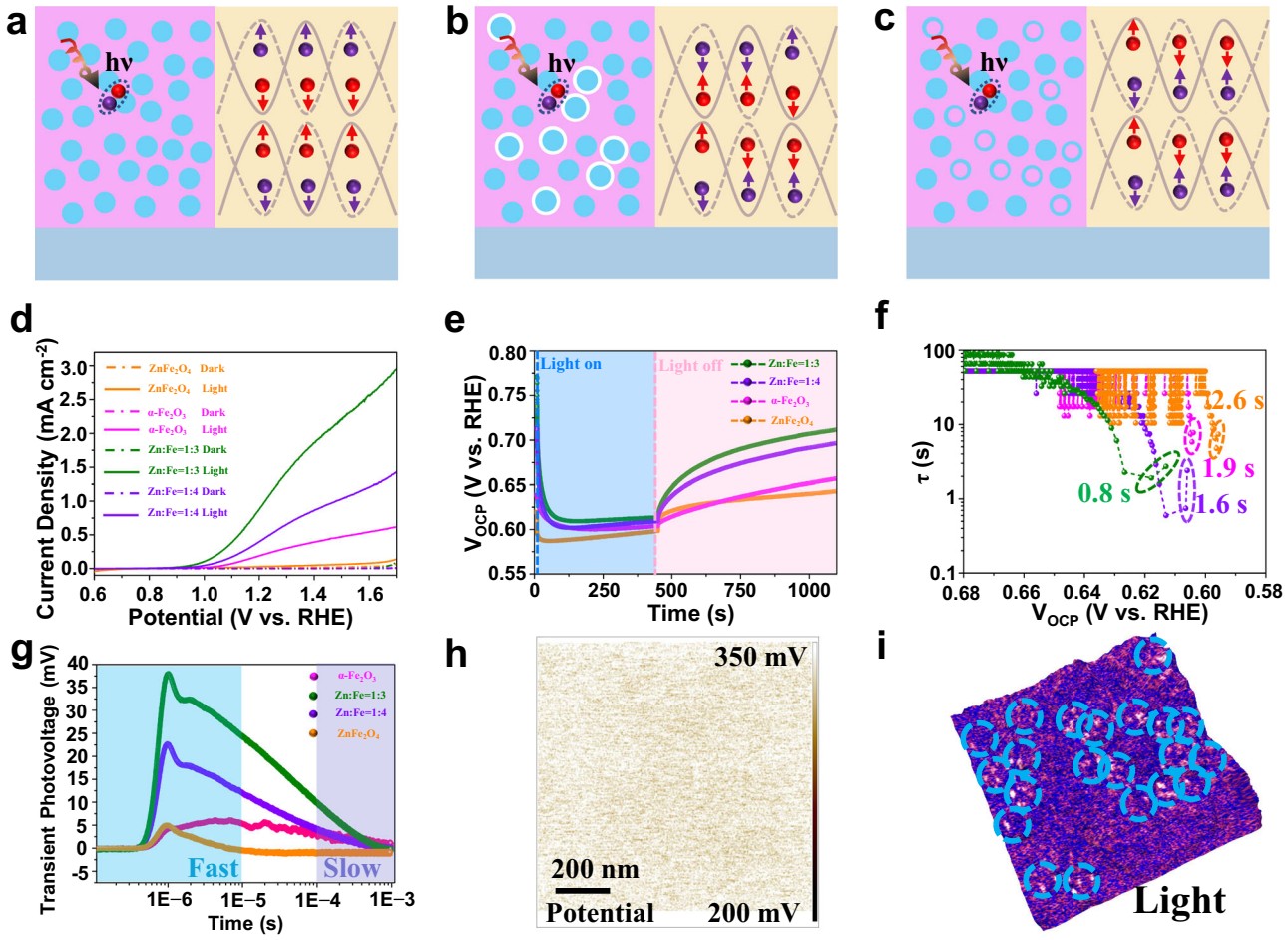

**Fig. 6 Enhancement mechanism of the Turing interface film. a** Schematic of the fundamental processes in the Turing interface film. Photon absorption and formation of the carriers and carrier separation at the interface. **b** Schematic of the fundamental processes in the conventional dual-phase interface film. Photon absorption and formation of the carriers and carrier separation at the interface. **c** Schematic of the fundamental processes in the non-Turing dual-phase interface film. Photon absorption and formation of the carriers and carrier separation at the interface. **d** Photocurrent density versus applied potential curves under 1 sun (AM 1.5 G, 100 mW cm$^{-2}$). **e** OCP curves for ZnFe$_2$O$_4$, the Turing interface film (Zn: Fe = 1:3), the Turing interface film (Zn: Fe = 1:4) and α-Fe$_2$O$_3$ films. The blue region represents the time in which the samples were under light illumination, and the pink region represents the time after the illumination was turned off. **f** OCP-derived carrier lifetimes. **g** Transient photovoltage responses (TPV). **h** V$_{CPD}$ image under 405 nm light irradiation of the Turing interface film (Zn: Fe = 1:3). **i** Current images (3D) of the Turing interface film (Zn: Fe = 1:3) overlaid under 405 nm light irradiation (1 V bias).

approximately zero at the interface. Therefore, based on the fewer dangling bonds (Fig. 5k), a strong bond between the ZnFe$_2$O$_4$ and α-Fe$_2$O$_3$ phases was exhibited to ensure a high level of interface contact and to enhance the built-in electric field.

**Photoelectrochemical measurement and charge separation enhancement mechanism.** Having identified the essential roles of the interface structure, we propose an interface transport mechanism that shows a large difference in the separation efficiency of photogenerated carriers. In the photoelectrode film, electron-hole pairs are generated under illumination, followed by different charge transport and trapping processes under bias. In the Turing interface film (Fig. 6a), the generated electron-hole pairs separate effectively at the interface via an enhanced interface built-in electric field. In the conventional dual-phase interface film (Fig. 6b), a built-in electric field at the interface is weakened due to the existence of the incoherent interface or semi-coherent interface. Therefore, the electron and hole pairs cannot be effectively separated at the interface, which increases the probability of recombination within the particle. In the case of a non-Turing dual-phase interface film (Fig. 6c), the built-in

electric field at the interface is also weakened, and the separated electrons and holes are recombined at the interface due to the mutual coating of the dual-phases. Above was further evaluated under AM 1.5 G irradiation with a standard three-electrode system in 1 M NaOH solution.

The Turing interface film (Zn: Fe = 1:3) exhibited a remarkable increase compared to the ZnFe$_2$O$_4$ and α-Fe$_2$O$_3$ films at 1.6 V vs. RHE, which reached 2.56 mA cm$^{-2}$ without photosensitizers and cocatalysts under 1 sun (100 mW cm$^{-2}$) (Fig. 6d). All of the Turing interface films showed improved performance under other light sources, including LEDs (Supplementary Fig. 30a) and Xe lamp (Supplementary Fig. 30b). The Turing interface film (Zn: Fe = 1:3) with an effective area of 1 cm$^2$ showed excellent stability (Supplementary Fig. 30c) at 1.23 V vs. RHE, indicating their photochemical durability for water splitting. The ratio of produced O$_2$ and H$_2$ was close to at the predicted 1:2 ratio, and the Faradaic efficiencies of the O$_2$ and H$_2$ evolution reactions were 93.1% and 95.8%, respectively (Supplementary Fig. 30d, e).

The charge separation and transfer were investigated by electrochemical measurements to understand the enhancement mechanism of the Turing interface films. Since the activation

energy of $SO_3^{2-}$ is low and the film is prone to photooxidation based on the kinetics, the *J-V* curve of the photoanode was measured with $Na_2SO_3$ as the hole scavenger. The $\eta_{sep}$ value of the Turing interface film (Zn: Fe = 1:3) is much higher than those of pure $ZnFe_2O_4$ and α-$Fe_2O_3$ films (Supplementary Fig. 30f), reaching 44.1% at 1.6 $V_{RHE}$, which was among the highest reported for iron-based films (Supplementary Table 1). The α-$Fe_2O_3$ film, the Turing interface film (Zn: Fe = 1:3) and the Turing interface film (Zn: Fe = 1:4) possessed a similar injection efficiency, which also revealed that the performance enhancement of the Turing structure came from the improvement of separation efficiency. The performance of various reference films has also been tested in detail (Supplementary Figs. 31 and 32). Moreover, the film demonstrated versatility for photoelectrochemical phenol oxidation (Supplementary Fig. 33). We sorted the photocurrent density of the samples at 1.6 V vs. RHE as a function of the α-$Fe_2O_3$ content. The performance of the conventional dual-phase interface films showed a linear change, while that of the Turing interface films showed a volcanic change (Supplementary Figs. 34 and 35a), which was related to the interface contact area. In addition, the optimized film thicknesses were kept at 300 nm, 300 nm and 3.2 μm for the Turing interface films, non-Turing dual-phase interface films and conventional dual-phase interface films (Supplementary Fig. 35b).

The band edge position was studied (Supplementary Figs. 36 and 37) to deeply understand the specific positions of the conduction band and valence band. Two nonpolar semiconductors may exhibit polar characteristics through interface design. The semi-coherent interface makes the acceptor action of the dangling bond form an upward band bending to achieve efficient carrier separation (see detailed information in Supplementary Fig. 38). The optical properties (Supplementary Fig. 39) of different samples were further evaluated. Photoelectrochemical impedance confirmed that the Turing interface film had the lowest interfacial resistance (Supplementary Fig. 40). To go a step closer to the real application, we scaled up the above-demonstrated high-efficiency, low-cost Turing structures film to spray a large-area 17 cm × 15 cm, which exhibited high-efficiency and green hydrogen production (Supplementary Fig. 41) and wastewater purification (Supplementary Fig. 42) driven by silicon solar cells (1.5 V) outdoors.

The open-circuit potential (OCP) transient decay profile provided additional information on the built-in electric field and its effects (Fig. 6e). The Turing interface film (Zn: Fe = 1:3) showed a remarkably accelerated OCP decay, indicative of larger photovoltage generation. In particular, band bending occurred, forming an internal electric field that separated the electron-hole pairs generated by the light, which were stored in the depletion zone. When the light illumination was terminated, the attenuation of the OCP accelerated. The carrier lifetime reached 0.8 s by including the Turing interface in the transient period when illumination was stopped (Fig. 6f), which is smaller by a factor of 3.2 compared to the value of ~2.6 s for $ZnFe_2O_4$. Fast decay kinetics were indicative of enhanced charge recombination when illumination was removed, which suggested that charge trapping was insignificant when the film was illuminated. The above result further confirmed the existence of an interface internal electric field. Consequently, effective charge separation was expected.

The surface photovoltage (SPV) of different specimens plotted as a function of wavelength, strongly demonstrated that the significant separation and transfer of photo-generated charges appeared in spatially with the aid of a strong intrinsic electric field (Supplementary Fig. 43). The dynamic properties of photoexcited charge separation and transfer processes were further explored by transient photovoltage technology (TPV) (Fig. 6g), which is widely used to understand the mechanism of semiconductor on the nanosecond time scale. Generally, the photo-generated charge separation process includes drift and diffusion: the drift process refers to the fast separation process ($<10^{-5}$ s) that occurs within the particles; the main diffusion process is the charge transfer between particles over a long period of time ($>10^{-4}$ s)[30]. Thus, the photovoltage response includes two parts: rising and decaying. The rising part of the photovoltage physically corresponds to the increase in the electron concentration of the conductive substrate of the electrode. Caused by the diffusion of photo-generated electrons to the substrate, the drop in photovoltage mainly corresponds to the recombination process of electrons leaving the conductive substrate. Therefore, charge transfer was tough in the bulk for $ZnFe_2O_4$ due to the fast recombination of photogenerated electrons and holes compared with the Turing interface film (Zn: Fe = 1:3). Moreover, an obvious peak was observed in the slow process owing to the strong interfacial electric field-dependent charge separation and transfer efficiency.

Spatial charge separation and transfer at the nanoscale were characterized with light-irradiated Kelvin probe force microscopy (KPFM). Upon light excitation, a distinct positive contact potential difference (CPD) arose in the Turing interface film (Zn: Fe = 1:3) (Fig. 6h and Supplementary Figs. 44a, b), meaning that more photogenerated holes had accumulated than the pristine α-$Fe_2O_3$ film (Supplementary Figs. 44c–e) on the surface. The respective current maps were simultaneously recorded at a bias voltage of 1 V (Supplementary Figs. 44f–h). The appearance of localized current bursts revealed the partial separation of highly efficient electron and hole pairs. The corresponding three-dimensional (3D) atomic force microscopy (AFM) topographical images were employed to identify the locations of the current burst sites (Fig. 6i and Supplementary Fig. 44i). More current bursts appeared after illumination, and it was concluded that the change was not caused by the film roughness. The Turing interface film (Zn: Fe = 1:3) displayed the lowest PL intensity and much longer lifetime compared with other samples (Supplementary Fig. 45).

To exclude the influence of the surface structure of the film, we conducted a surface roughness measurement. According to the AFM micrographs (2D and 3D) in Supplementary Figs. 46 and 47, there is little difference between the Turing structure film and the non-Turing structure film, and the surface roughness is distributed at 5–8 nm. Therefore, we can be sure that the same preparation process has little effect on the surface roughness of the film. Moreover, there is no strong dependence between its performance and surface roughness. We confirm that the enhancement in the efficiency of the thin film is attributed to the formation of the bulk Turing interface. In order to further verify that the formation of the Turing structure is determined by the intrinsic property of the solution rather than an artifact that can appear due to the fabrication process, we used the drop coating and spin coating technique to prepare high-performance Turing structure films (see detailed information in Supplementary Figs. 48 and 49).

In addition, to eliminate the interference of α-$Fe_2O_3$ and verify the universal applicability of our strategy, Zn: Fe = 1:2.4, Zn: Fe = 1:6 (Supplementary Fig. 50), Zn-doped α-$Fe_2O_3$, $CuFe_2O_4$, and $MgFe_2O_4$ films were also meticulously considered (Supplementary Figs. 51–55). The photocurrent density (AM 1.5 G, 100 mW cm$^{-2}$) at 1.6 V vs. RHE depending on the α-$Fe_2O_3$ content are shown (Supplementary Fig. 56). As expected, the variation trend was very similar, which also verified the universality of the strategy of constructing atomic-level Turing interfaces. We determined that the Turing interface can vastly enhance charge separation efficiency, which provided a new perspective for designing an excellent performance film.

## Discussion

Our study demonstrates that a Turing structure with a near zero strain semi-coherent interface can be realized for an inorganic semiconductor thin film system by re-coordination in homogeneous solutions. Additionally, the Turing interface at the atomic level was revealed. Notably, accurate composition and performance analyses were performed to validate our designs, which were shown to accelerate the separation of photogenerated electron-hole pairs and reduce the bulk recombination probability. The promising large-area film showed the great potential application prospects for spinel ferrite film. This strategy may pave a new path to solve the problem of charge recombination in energy conversion.

## Methods

**Fabrication of atomic-level Turing interface films (Zn).** First, 0.0045 mol zinc acetate and 0.00135 mol iron (III) 2,4-pentanedionate (Zn: Fe = 1:3) were dissolved in 90 mL of methanol solution. After 10 min of ultrasonication, the solution was sprayed onto fluorine-doped $SnO_2$ (FTO) glass substrates with the help of a homemade spray pyrolysis system composed of a spray head and a heating table. Thermocouples were used to monitor the temperature of FTO, which was maintained at 400 °C during the spraying process, and the spraying time was ~45 min. Then, the substrates were transferred to a muffle furnace and calcined in air at 600 °C for 1 h and then at 700 °C for 10 min according to the original heating rate. For the Zn: Fe = 1: $x$ photoelectrodes, a similar synthesis process was carried out without the addition of an iron precursor.

**Fabrication of atomic-level Turing interface films (Cu and Mg).** The zinc acetate in the above scheme was replaced with copper acetate and magnesium acetate to prepare Cu: Fe = 1: $x$ and Mg: Fe = 1: $x$ photoelectrodes, respectively.

**Fabrication of Zn-doped $\alpha$-$Fe_2O_3$ films.** Here, 0.009 mol iron (III) 2,4-pentanedionate was dissolved in 90 mL of methanol solution, and the amount of zinc acetate (1%, 3%, 5%, 7%) was determined by the amount of iron. The subsequent fabrication steps, including the spraying, heating, and transferring processes, were consistent with the preparation process mentioned earlier.

**Fabrication of $ZnFe_2O_4$ and $\alpha$-$Fe_2O_3$ powder.** The preparation process of the powder was similar to that of the films, although the methanol solution was put into an oven to be dried and then put into a muffle furnace to be calcined in the same pattern. The chemical reactions involved are as follows:

$$ZnO + \alpha - Fe_2O_3 = ZnFe_2O_4 \quad (2)$$

$$ZnO + 3/2\,\alpha - Fe_2O_3 = ZnFe_2O_4 + 1/2\,\alpha - Fe_2O_3 \quad (3)$$

$$ZnO + 2\,\alpha - Fe_2O_3 = ZnFe_2O_4 + \alpha - Fe_2O_3 \quad (4)$$

**Preparation of the conventional dual-phase interface films of $ZnFe_2O_4$ and $\alpha$-$Fe_2O_3$.** Films with different proportions (Supplementary Table 2) were prepared by electrophoretic deposition on FTO according to our research group[31]. Iodine (10 mg) and sample (50 mg) were suspended in 40 mL acetone after being dispersed by sonication. Particles are deposited on the FTO substrate at a bias of 30 V for 2 min. Then use $TiCl_4$ methanol solution for multiple necking treatment.

**Preparation of the non-Turing dual-phase interface films of $ZnFe_2O_4$ and $\alpha$-$Fe_2O_3$.** The preparation is similar to that of atomic-level Turing interface films (Zn); only $FeCl_3$ and $Fe(NO_3)_3$ are used instead of $Fe[C_5H_7O_2]_3$.

**Photo-deposition of Au and $MnO_x$.** Photo-deposition was performed following the reported procedure[32] with minor modifications. First, 30 mg of the sample was suspended in 60 mL of deionized water and methanol solution. Then, 0.2 mL of $HAuCl_4$ solution (5 mg mL$^{-1}$) was added, and the suspension was irradiated with a 300 W Xe lamp ($\lambda > 420$ nm) for 3 h. Photodeposition of $MnO_x$ was achieved with 30 mg of the sample suspended in 50 mL of deionized water containing 1.6 g of $NaIO_3$. Then, 1 mL of $MnSO_4$ solution (2 mg mL$^{-1}$) was added, and the suspension was irradiated with a 300 W Xe lamp ($\lambda > 420$ nm) for 3 h.

**Characterization.** X-ray diffraction (XRD) analysis was carried out using a Rigaku Ultima III X-ray diffractometer with Cu Kα radiation (40 kV, 40 mA). The morphologies and microstructures of all the samples were examined by a Zeiss Gemini SEM 500 ultrahigh resolution field-emission scanning electron microscope equipped with an Oxford energy-dispersive X-ray spectroscopy (EDS) system and by transmission electron microscopy (TEM, JEOL JEM-2100). To further

accurately analyze the microstructure of the films, a focused ion beam (ZEISS Crossbeam 540) was used for sample preparation. High-angle annular dark-field scanning transmission electron microscopy (HAADF-STEM) images were taken using a Titan Cubed Themis G2 300. Reflection and transmission spectra were measured on a Lambda 950 (Perkin Elmer) spectrometer equipped with a lab-sphere diffuse reflectance accessory. The shape evolution of the droplets was observed through a microscope (IX71, Olympus, Japan). The binding energy and valence band spectra of the films were measured by X-ray photoelectron spectroscopy (XPS, UIVAC-PHI, Japan). Fourier transform infrared spectra (FT-IR) of the samples were obtained from NEXUS870, scanning from 4000 to 400 cm$^{-1}$ in KBr tablets. The surface roughness and thickness of the films were tested by an optical interference profilometer (Contour GT-K1, VECCO). Thermogravimetry (TG) and differential scanning calorimetry (DSC) were performed with the Netzsch STA449F3, when compressed air and nitrogen were used as the carrier gas, and an alumina crucible was used to hold the samples. The heating rate was 10 °C min$^{-1}$. Photoluminescence (PL) and time-resolved PL (TRPL) measurements were performed on the films using a Horiba Fluorolog3 spectrofluorometer. The excitation wavelength was 480 nm, and the probed wavelength was 530 nm. The TRPL curves were fitted by a double-exponential function. The lock-in-based surface photovoltage (SPV) spectroscopic (CEL-SPS1000) measurement system consisted of a source of monochromatic light, a sample cell, a computer, and a lock-in amplifier (SR830) with a light chopper (SR540). A low chopping frequency of 170 Hz was used. A 150 W xenon lamp (Osram bulb) and a grating monochromator provided monochromatic light. Transient photovoltage (TPV, CEL-TPV2000) measurements were performed to study the behaviors of the photogenerated charge carriers in the system. The samples were excited with a laser radiation pulse with a wavelength of 355 nm and a pulse width of 5 ns from a third-harmonic Nd: YAG laser. The TPV signals were recorded by a digital phosphor oscilloscope. The diffusion concentration of metal ions was determined based on the spectrum obtained by an inductively coupled plasma spectrometer (ICP, Avio500). The detailed surface roughness of the film was measured by atomic force microscopy (AFM, Bruker Dimension Icon, Germany).

**Atom probe tomography test.** To prepare the appropriate samples for the APT test, we used magnetron sputtering to plate a layer of metallic aluminum on the surface of the film that prevented damage to the sample during the test. The APT experiments were conducted on a CAMECA LEAP 5000 XR instrument equipped with an ultraviolet laser with a spot size of 2 μm and a wavelength of 355 nm. The detection efficiency of this state-of-the-art microscope was ~52%. Data were acquired in the laser pulsing mode at a specimen temperature of 60 K with a target evaporation rate of 5 ions/1000 pulses, a pulse rate of 200 kHz, and a laser pulse energy of 120 pJ. The APT data were reconstructed and analyzed using commercial IVAS 3.8.4 software.

**Kelvin probe force microscopy.** Surface potential (contact potential difference, CPD) images and signals of the samples were measured using KPFM (Bruker) under ambient atmospheric conditions in amplitude-modulated (AM-KPFM) mode. The CPD is equal to the work function of the tip minus the work function of the sample. During the measurement of the surface potential, lift mode was used with a lift height of 20 nm considering the signal-to-noise ratio. In lift mode, the topography and the surface potential signals were sequentially recorded. A Pt/Ir-coated Si tip was used as the Kelvin tip with a spring constant of 1–5 N m$^{-1}$ and a resonant frequency of 60–100 kHz. To measure the surface potential under illumination, a 405 nm laser with a neutral density filter was used. The light intensity could be adjusted from 0 to 2 mW cm$^{-2}$, ensuring saturation of the measured surface potential under illumination.

**Conductive atomic force microscopy.** The conductive atomic force microscopy (C-AFM) images were obtained by conductive AFM (Peakforce TUNA, Bruker). Pt/Ir-coated conductive AFM tips were applied to achieve electrical contact with the surface of electrodes. A 1 V bias was applied on the AFM tip during the measurement. The light conditions were the same as those in the KPFM measurements ($\lambda = 405$ nm, light intensity = 2 mW cm$^{-2}$).

**Photoelectrochemical water splitting.** The photoelectrochemical performance of the photoelectrodes with an area of 1 cm$^2$ without a mask was measured in a three-electrode cell with a quartz window using an electrochemical analyser (CHI-760E, Shanghai Chenhua) under AM 1.5 G illumination with an intensity of 100 mW cm$^{-2}$. AM 1.5 G simulated sunlight was obtained from a Newport Sol3A Class AAA simulator, which was calibrated at 100 mW cm$^{-2}$ by a Newport silicon cell. LED illumination (LEDGUHON, China) was used to compare the improved performance under different light intensities. The spotlight experiment (500 mW cm$^{-2}$) was performed with a xenon lamp source (Beijing Perfectlight, PLS-SXE 300DUV) equipped with a convex lens. The light intensity was calibrated by using a certified reference cell. The photocurrent under different light intensities was measured, where the light intensity was measured by an optical power meter (PM100D, Thorlabs). The fabricated electrode, Pt foil and a Ag/AgCl electrode were used as the working, counter, and reference electrodes, respectively. A 1 M NaOH aqueous solution (pH = 13.6) was employed as the electrolyte. The reversible hydrogen electrode (RHE) potential

was calculated according to the Nernst equation: $V_{RHE} = V_{Ag/AgCl} + 0.059 \text{ pH} + E^0_{Ag/AgCl}$, where $E^0_{Ag/AgCl} = 0.1976$ V at 25 °C. The J–V curves were measured at 30 mV s$^{-1}$ in the range from −0.4 to +0.7 V vs. Ag/AgCl. To verify the stability of the photoelectrodes and the photoelectrochemical conversion efficiency, i-t curves were obtained. The photoelectrochemical cell was sealed and purged with nitrogen for a long time to clear the air, and the generated gas was detected by gas chromatography (Shimadzu, GC-8A) to calculate the Faraday efficiency. The detailed formulas are as follows:

$$n_{theory}(H_2) = Q/2F \quad \eta(H_2) = n(H_2)/n_{theory}(H_2) \tag{5}$$

$$n_{theory}(O_2) = Q/4F \quad \eta(O_2) = n(O_2)/n_{theory}(O_2) \tag{6}$$

where $n_{theory}(H_2)$ and $n_{theory}(O_2)$ represent the amount of hydrogen and the amount of oxygen theoretically produced, respectively; $n(H_2)$ and $n(O_2)$ represent the actual amounts of hydrogen and oxygen, respectively; Q is the amount of electricity generated in the circuit during illumination; and F is the Faraday constant, $F = 96485$ C mol$^{-1}$. Mott-Schottky plots were employed to determine the flat band potential of semiconductors at alternating current (AC) frequencies of 0.1 kHz, 0.5 kHz, and 1 kHz.

The charge separation efficiency ($\eta_{sep}$) and charge injection efficiency ($\eta_{inj}$) were calculated using Eqs. (7) and (8), respectively.

$$\eta_{sep} = J_{Na_2SO_3}/J_{abs} \tag{7}$$

$$\eta_{inj} = J_{H_2O}/J_{Na_2SO_3} \tag{8}$$

where $J_{H_2O}$ represents the photocurrent density without a hole scavenger and $J_{Na_2SO_3}$ is the photocurrent density with the addition of a $Na_2SO_3$ hole scavenger. $J_{abs}$ was determined by integrating the UV-vis absorption spectra of thin films with respect to the AM 1.5 G solar light spectrum. Photoelectrochemical impedance spectroscopic (PEIS) curves were measured by an electrochemical analyser (Solartron 1260+1287) under a forward bias of 1.6 $V_{RHE}$ and a 300 W Xenon lamp with an AM 1.5 G filter. The frequency ranged from 0.1 Hz to 1000 kHz.

**Photoelectrochemical pollutant disposal**. Electrocatalytic (EC), photocatalytic (PC) and PEC degradation tests were studied using a three-electrode system under AM 1.5 G illumination. The reactive system contained a mixture solution of 100 mL $Na_2SO_4$ (0.1 M) and 20 mg L$^{-1}$ phenol. The dispersed solution was magnetically stirred in the dark for 2 h to ensure absorption-desorption equilibrium. The concentration of phenol was measured by the 4-aminoantipyrine method according to our previous report[33].

**Diffusion coefficient measurement**. A diaphragm cell with upper and lower compartments, which were separated by a porous membrane, was used to measure the diffusion coefficients[22,34]. The initial concentration of the solution is shown (Supplementary Table 3). The diffusion coefficient (D) was calculated according to the following formulas:

$$D = \frac{\ln\frac{c_1-c_2}{c_3-c_4}}{\beta t} \tag{9}$$

$$M = \ln\frac{c_1-c_2}{c_3-c_4} \tag{10}$$

$$N = \ln\frac{c_1-c_2}{c_3-c_4} \tag{11}$$

$c_1$ and $c_3$ are the initial and final concentrations in the lower compartment, respectively, $c_2$ and $c_4$ are the initial and final concentrations in the upper compartment, respectively, $\beta$ is the cell constant, and t is the time.

Diffusion coefficient measurements were also carried out by nuclear magnetic resonance spectrometry (NMR, 600 M, Bruker).

**Computational details**. The band structure in this study was determined using the Vienna ab initio simulation package (VASP)[35,36], which employs the spin-polarized DFT method. The generalized gradient approximation (GGA)[37] in the Perdew-Burke-Ernzerhof (PBE) scheme[38] was used for the exchange correlation functional. The projector augmented wave (PAW)[39] pseudopotential was used to describe the core and valence electrons. The strong on-site Coulomb repulsion between localized 3d electrons was modified by considering the Hubbard U correction proposed by Dudarev et al., and an effective U value of 5.3 eV was used for Fe 3d orbitals[40]. The cut-off energy was 520 eV, which is high enough to ensure that no Pulay stresses occur within the cell during geometric relaxations in all materials. In addition, 5 × 5 × 1, 2 × 2 × 2, 2 × 2 × 2, and 2 × 2 × 2 k-point samplings in reciprocal space were used for α-Fe₂O₃, MgFe₂O₄, CuFe₂O₄, and ZnFe₂O₄, respectively. Geometric relaxations were carried out until the residual forces on each ion converged to be <2 × 10$^{-5}$ eV Å$^{-1}$.

The DFT calculations were performed using Gaussian16[41] at the B3LYP[42]/6-311 G(d,p)[43,44] level with the D3(BJ) empirical dispersion correction[45]. We took the solvent effects of methanol into account by using the SMD model[46]. Gauss

View can create a surface where the color is determined by the values of a second property, which implies that we can map the values of one property on an iso-surface of a different property. The cubegen (a module of Gaussian software) was first employed to calculate the electron density iso-surface (0.001 e bohr$^{-3}$). Subsequently, the electrostatic potential value of the point on the iso-surface was calculated and displayed on the electron iso-surface by color. Then, according to the electrostatic potential value of different molecules, the electron density surface was drawn.

For MD simulations of C solution (Fe(NO₃)₃, Zn(CH₃COO)₂), the system was modeled by a cubic cell with 30 $Zn^{2+}$, 90 $Fe^{3+}$, 330 $NO_3^-$, and 500 methanol molecules. For MD simulations of F solution (Fe[C₅H₇O₂]₃, Zn(CH₃COO)₂), the system was modeled by a cubic cell with 30 $Zn^{2+}$, 90 $Fe^{3+}$, 270 $CH_3COO^-$, 60 [C₅H₇O₂]$^-$, and 500 methanol molecules. After geometry optimization, the simulations were performed in an NPT ensemble (298 K) with a total simulation time of 10 ns at a time step of 1 fs.

**Calculation formulas involved**.

$$J_p = J_{abs} \times \eta_{inj} \times \eta_{sep} \tag{12}$$

where $J_p$ represents the theoretical photocurrent density, $\eta_{inj}$ is the charge injection efficiency, and $\eta_{sep}$ is the charge separation efficiency.

The M-S equation is as follows[47]:

$$\frac{1}{C^2} = \frac{2\left(V - V_{fb} - \frac{K_b T}{e}\right)}{\varepsilon\varepsilon_0 A^2 e N_D} \tag{13}$$

where e is the elementary charge, $\varepsilon$ is the dielectric constant, $\varepsilon_0$ is the permittivity of vacuum, $N_D$ is the donor density, C is the space charge capacitance of the semiconductor, A is the electrode surface area, V is the potential applied at the film, $V_{fb}$ is the flat band potential, $K_b$ is Boltzmann's constant, and T is the absolute temperature.

The absorption coefficient $\alpha$ as a function of wavelength $\lambda$ was determined by spectrophotometry from the special absorption[48]:

$$\alpha(\lambda) = \frac{\ln\frac{1-R(\lambda)}{T(\lambda)}}{d} \tag{14}$$

where d is the film thickness and $R(\lambda)$ and $T(\lambda)$ are the reflection and transmission at specific wavelengths, respectively.

The lifetime of the generated charges as a function of OCP was calculated according to the following equation:

$$\tau_n = \frac{k_B T\left(\frac{dOCP}{dt}\right)^{-1}}{e} \tag{15}$$

in which $\tau_n$ is the potential-dependent carrier lifetime, $k_B$ is Boltzmann's constant, T is the temperature in K, e is the charge of a single electron, and $\frac{dOCP}{dt}$ is the derivative of the OCP transient decay.

**Reporting summary**. Further information on research design is available in the Nature Research Reporting Summary linked to this article.

## Data availability

The data that support the findings of this study are available from the corresponding authors upon reasonable request.

## Code availability

Numerical simulations in this work are all performed using the commercial finite element software MATLAB. All related codes can be built with the instructions in the Supplementary Information and available from the corresponding authors upon reasonable request.

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

## Acknowledgements

This work was primarily supported by the National Science Fund for Distinguished Young Scholars (22025202), National Key Research and Development Program of China (No. 2018YFA0209303), National Natural Science Foundation of China (No. 51972165), and Natural Science Foundation of Jiangsu Province of China (No. BK20202003). The numerical calculations in this Letter have been done on the computing facilities in the High Performance Computing Center (HPCC) of Nanjing University.

## Author contributions

Z.L. conceived the idea and directed the project. Y.Z. carried out the synthesis and characterization of the samples and wrote the manuscript. N.Z., Y.C., H.H., and Y.L. assisted in data analysis. X.X. employed magnetron sputtering to plate aluminum on the surface of the film. Y.L. and F.F. performed KPFM and C-AFM measurements. W.W. performed the density functional theory calculation. J.Y. and Z.Z. discussed the results and improved the paper. All authors commented on the manuscript.

## Competing interests

The authors declare no competing interests.
