## [Peer Review File · Nature Communications]

Homogeneous solution assembled Turing structures with near zero strain semi-coherence interfaceREVIEWER COMMENTS

Reviewer #1 (Remarks to the Author):

In the manuscript "Homogeneous solution assembled Turing structures with near zero strain semi-coherence interface," Zhang et al. reported that the Turing structure can be realized for an inorganic semiconductor thin film system by re-coordination in homogeneous solutions. The authors concluded that the Turing interface can enhance charge separation efficiency.

The experimental data are technically sound, with various measurements shown in Supplementary Information (SI). Nevertheless, it must be said that this paper has serious flaws.

The authors claim that the pattern in Fig. 1c is the Turing pattern. However, no clear evidence supports this claim. The authors must provide evidence of the Turing pattern in sufficient quality and quantity. It is well known in mathematical biology that one cannot claim the Turing pattern just because it looks like the Turing pattern. If the authors insist on claiming that their interface film is the Turing pattern, they should at least show that the same pattern can be derived from Turing's reaction-diffusion equation. In addition to this, the following statements are overstatements because there is not enough evidence to support their proposal:
"we ... proposed a new concept of a Turing interface at the atomic level"
"a new concept of a Turing interface at the atomic level was presented for the first time,"

From the viewpoint of readability, the manuscript is not well written. They cite SI too often, preventing readers from understanding the essence of the paper. The main text itself should powerfully state the author's views.

For these reasons, I cannot recommend this manuscript for publication in Nature Communications.

Reviewer #2 (Remarks to the Author):

This manuscript (NCOMMS- 21-39616) describes the synthesis of near-zero strain semi-coherent Turing interface spinel ferrites films with various Zn:Fe, Cu:Fe and Mn:Fe ratios with enhanced photoelectrochemical activities. This work provides the details of the synthesis of Turing interface spinel ferrites film and its advantages and related characterization for solar water splitting and photoelectrochemical phenol oxidation for the first time. This work elaborates a new insight and clear understanding of using Turing interface spinel ferrites films for energy-related applications and related areas. Hence, considering the detailed systematic studies carried out in this work, well supported by experiments and theoretical studies, this manuscript may be considered for publication in Nature Comm after the appropriate revisions as mentioned below. The authors should include and address the following points in the revised manuscript clearly.

1. How can one confirm that the Turing structure is formed? Is there any direct study/simulation etc. to support the claim.
2. How the images for Turing structure were developed from the SEM images is not clear. For example, for Supplementary Fig. 4. The authors should include details in the revised manuscript.
3. In Supplementary Fig. 4, the presence of α -Fe₂O₃ and ZnFe₂O₄ phases should be pointed clearly.
4. The information about the LED light used in this manuscript is missing. What is the wavelength of LED used?
5. How the authors calculated the 'molecular electrostatic potential' of the surfaces is not provided in Supplementary information. The authors should provide the details.
6. Supplementary Fig. 28f is not clear and difficult to understand.
7. What is the area or the volume of the electrode materials used for the study of the amount of O₂ and H₂ produced should be included. For example, in Fig. Supplementary Fig. 30 b, c and d and other such plots in the whole manuscript in SI.
8. I do not understand the necessity to study the magnetic property of the material. Can the authors explain the same?

Reviewer #3 (Remarks to the Author):

It is a nice piece of work on Interfacial engineering which would be useful for many energy applications. The work is interesting and can be published after incorporating these comments.

1. Though the role of Turing structures is known, authors should include a paragraph on the mechanism and role of such Turing structures in improving the charge recombination.
2. What is the role of FeCl_3 and $\text{Fe}(\text{NO}_3)_3$ during the film formation process.
3. What is the reason of high output in the case of Turing interface film (Zn: Fe=1:3)
4. An explanation to be added to why charge transfer is non-existent in bulk of Zn ferrite
5. Can this be applied to perovskite solar cells as interfacial properties play a major role in the same.
6. What is the surface roughness of the film and did it have any influence on the efficiency.
7. Do the fabrication of these films have a role in deciding the Turing structures.
8. What is the thickness of the films.
9. Similar reports should be compared in the discussion.

REVIEWERS COMMENTS

Reviewer #1 (Remarks to the Author):

In the manuscript "Homogeneous solution assembled Turing structures with near zero strain semi-coherence interface," Zhang et al. reported that the Turing structure can be realized for an inorganic semiconductor thin film system by re-coordination in homogeneous solutions. The authors concluded that the Turing interface can enhance charge separation efficiency.

The experimental data are technically sound, with various measurements shown in Supplementary Information (SI). Nevertheless, it must be said that this paper has serious flaws.

The authors claim that the pattern in Fig. 1c is the Turing pattern. However, no clear evidence supports this claim. The authors must provide evidence of the Turing pattern in sufficient quality and quantity. It is well known in mathematical biology that one cannot claim the Turing pattern just because it looks like the Turing pattern. If the authors insist on claiming that their interface film is the Turing pattern, they should at least show that the same pattern can be derived from Turing's reaction-diffusion equation. In addition to this, the following statements are overstatements because there is not enough evidence to support their proposal:

"we ... proposed a new concept of a Turing interface at the atomic level"

"a new concept of a Turing interface at the atomic level was presented for the first time,"

From the viewpoint of readability, the manuscript is not well written. They cite SI too often, preventing readers from understanding the essence of the paper. The main text itself should powerfully state the author's views.

For these reasons, I cannot recommend this manuscript for publication in Nature Communications.

Reviewer #2 (Remarks to the Author):

This manuscript (NCOMMS-21-39616) describes the synthesis of near-zero strain semi-coherent Turing interface spinel ferrites films with various Zn: Fe, Cu: Fe and Mg: Fe ratios with enhanced photoelectrochemical activities. This work provides the details of the synthesis of Turing interface spinel ferrites film and its advantages and related characterization for solar water splitting and photoelectrochemical phenol oxidation for the first time. This work elaborates a new insight and clear understanding of using Turing interface spinel ferrites films for energy-related applications and related areas. Hence, considering the detailed systematic studies carried out in this work, well supported by experiments and theoretical studies, this manuscript may be considered for

publication in Nature Comm after the appropriate revisions as mentioned below. The authors should include and address the following points in the revised manuscript clearly.

1. How can one confirm that the Turing structure is formed? Is there any direct study/simulation etc. to support the claim.
2. How the images for Turing structure were developed from the SEM images is not clear. For example, for Supplementary Fig. 4. The authors should include details in the revised manuscript.
3. In Supplementary Fig. 4, the presence of α -Fe₂O₃ and ZnFe₂O₄ phases should be pointed clearly.
4. The information about the LED light used in this manuscript is missing. What is the wavelength of LED used?
5. How the authors calculated the 'molecular electrostatic potential' of the surfaces is not provided in Supplementary information. The authors should provide the details.
6. Supplementary Fig. 28f is not clear and difficult to understand.
7. What is the area or the volume of the electrode materials used for the study of the amount of O₂ and H₂ produced should be included. For example, in Fig. Supplementary Fig. 30 b, c and d and other such plots in the whole manuscript in SI.
8. I do not understand the necessity to study the magnetic property of the material. Can the authors explain the same?

Reviewer #3 (Remarks to the Author):

It is a nice piece of work on Interfacial engineering which would be useful for many energy applications. The work is interesting and can be published after incorporating these comments.

1. Though the role of Turing structures is known, authors should include a paragraph on the mechanism and role of such Turing structures in improving the charge recombination.
2. What is the role of FeCl₃ and Fe(NO₃)₃ during the film formation process.
3. What is the reason of high output in the case of Turing interface film (Zn: Fe=1:3)
4. An explanation to be added to why charge transfer is not existent in bulk of Zn ferrite.
5. Can this be applied to perovskite solar cells as interfacial properties play a major role in the same.
6. What is the surface roughness of the film and did it have any influence on the efficiency.
7. Do the fabrication of these films have a role in deciding the Turing structures.
8. What is the thickness of the films.
9. Similar reports should be compared in the discussion.

Manuscript Type: Article

Title: Homogeneous solution assembled Turing structures with near zero strain semi-coherence interface.

Author(s): Yuanming Zhang, Ningsi Zhang, Yong Liu, Yong Chen, Huiting Huang, Wenjing Wang, Xiaoming Xu, Yang Li, Fengtao Fan, Jinhua Ye, Zhaosheng Li, Zhigang Zou

General response: We sincerely thank the editor, editorial staff and all reviewers for their critical comments that we have based on to improve the quality of our manuscript. The manuscript has been modified point-by-point after addressing all the suggestions as listed below.

(Our response is given in blue, some key sentences are highlighted in yellow and the corrections in the revised manuscript are shown in red)

Points-by-points responses to Reviewer(s)

Reviewer #1 (Remarks to the Author):

In the manuscript "Homogeneous solution assembled Turing structures with near zero strain semi-coherence interface," Zhang et al. reported that the Turing structure can be realized for an inorganic semiconductor thin film system by re-coordination in homogeneous solutions. The authors concluded that the Turing interface can enhance charge separation efficiency.

The experimental data are technically sound, with various measurements shown in Supplementary Information (SI). Nevertheless, it must be said that this paper has serious flaws.

Response: We are very grateful for the reviewer. The review comments are very constructive to further improve the quality of the manuscript. We have addressed the comments point-by-point and made the corresponding changes accordingly in the revised manuscript. We hope our revision with new experimental results and the expanded theoretical simulations has convinced the reviewer of the significance and the originality of our work.

The authors claim that the pattern in Fig. 1c is the Turing pattern. However, no clear evidence supports this claim. The authors must provide evidence of the Turing pattern in sufficient quality and quantity. It is well known in mathematical biology that one cannot claim the Turing pattern just because it looks like the Turing pattern. If the authors insist on claiming that their interface film is the Turing pattern, they should at least show that the same pattern can be derived from

Turing's reaction-diffusion equation.

Response: We thank the reviewer for insightful and constructive suggestions. As the reviewer pointed out, one cannot claim the Turing pattern just because it looks like the Turing pattern. According to the suggestion, we conducted the experiments and simulations, including two-dimensional diffusion-ordered nuclear magnetic resonance spectroscopy (2D DOSY) (Fig. 3a), molecular dynamics (MD) simulation (Fig. 3b-d) and numerical simulation (Fig. 3h and Supplementary Fig. 19). We will analyze these results in detail below.

Fig. 3. Understanding the Turing structure by the reaction-diffusion equation. **a**, Two-dimensional diffusion-ordered nuclear magnetic resonance spectroscopy (2D DOSY) of a mixed solution of $\text{Fe}[\text{C}_5\text{H}_7\text{O}_2]_3$ and $\text{Zn}(\text{CH}_3\text{COO})_2$. **b**, Mean square displacement of Zn^{2+} in different solutions in Fig. 2f. The purple red and dark green curves represent the C solution ($\text{Fe}(\text{NO}_3)_3$, $\text{Zn}(\text{CH}_3\text{COO})_2$) and F solution ($\text{Fe}[\text{C}_5\text{H}_7\text{O}_2]_3$, $\text{Zn}(\text{CH}_3\text{COO})_2$), respectively. **c**, Snapshots of the dynamic process of F solution ($\text{Fe}[\text{C}_5\text{H}_7\text{O}_2]_3$, $\text{Zn}(\text{CH}_3\text{COO})_2$) at different

simulation times. The orange balls are Zn^{2+} , the green balls are Fe^{3+} , the pink represents $[C_5H_7O_2]^-$ and the cyan represents CH_3COO^- . **d**, Snapshots of the dynamic process of C solution ($Fe(NO_3)_3$, $Zn(CH_3COO)_2$) at different simulation times. The orange balls are Zn^{2+} , the green balls are Fe^{3+} , and purple red represents NO_3^- . **e**, SEM image of Zn: Fe=1:2.4 film. **f**, the Turing interface film (Zn: Fe=1:3.5). **g**, the Turing interface film (Zn: Fe=1:4). In the fourth row of the figure, blue represents $ZnFe_2O_4$, and pink represents $\alpha-Fe_2O_3$. **h**, Patterns exhibited by prey and predator of the system at $T = 100, 600$ and 1000 , respectively. The difference in diffusion coefficient is 30 times. Scale bars: **c, d, e**, 100 nm.

Supplementary Fig. 19. Numerical simulation. **a**, Patterns exhibited by prey and predator of the system at $T = 1, 100, 600$ and 1000 , respectively. $d = \frac{1}{30}$, and the initial data is in line with the equation 13. **b**, Patterns exhibited by prey and predator of the system at $T = 1, 100, 600$ and 1000 , respectively. $d = \frac{1}{20}$, and the initial data is in line with the equation 14.

Turing structures typically emerge in reaction-diffusion processes far from thermodynamic equilibrium, involving at least two chemicals with different diffusion coefficients (inhibitors and activators) [r_1, r_2]. Alan Turing mathematically showed that the stable state can destabilize under certain conditions and spontaneously generate spatially stationary patterns in a reaction-diffusion system, which is particularly well known in mathematical biology. Thus, the Turing pattern can be described by dimensionless reaction-diffusion equations combined with our chemical system:

$$\begin{cases} \frac{\partial u}{\partial t} = f(u, v) + d_u \Delta u \\ \frac{\partial v}{\partial t} = g(u, v) + d_v \Delta v \end{cases}$$

where u and v is the vector of concentration, the functions $f(u, v)$ and $g(u, v)$ represent the reaction kinetics (diffusion term), and d_u and d_v are the diffusion coefficients of u and v , respectively. The generation of the Turing pattern corresponds to the coupling of a nonlinear reaction kinetic process and a special diffusion process, which will be unstable due to the different diffusion velocities of the two factors.

First, we verified the activators and inhibitors with big difference in diffusion coefficient in homogeneous solution by the 2D DOSY and MD simulation. As shown in Fig. 3a, it should be noted that the diffusion coefficient of $(CH_3COO)^-$ was 4 times higher than the diffusion coefficient

of $[\text{C}_5\text{H}_7\text{O}_2]^-$ in deuterated methanol for the 2D DOSY test due to the difference in molecular weight. Next, to gain an atomic-level understanding of the re-coordination mechanism in homogeneous solutions, MD simulations were explored. There is an order of magnitude difference in the diffusion coefficients of Zn^{2+} from $\text{Zn}(\text{CH}_3\text{COO})_2$ in the $\text{Fe}(\text{NO}_3)_3$ and $\text{Fe}[\text{C}_5\text{H}_7\text{O}_2]_3$ solutions (Fig. 3b). Meanwhile, it is clear that Zn^{2+} and $[\text{C}_5\text{H}_7\text{O}_2]^-$ show much higher affinity than CH_3COO^- (Fig. 3c), meaning that the dynamic process of Zn^{2+} is slow. While the movement of Zn^{2+} and Fe^{3+} was not affected by NO_3^- (Fig. 3d). The above experiments and MD simulations results provide solid evidence that high molecular weight organic anions play a key role in homogeneous solution for formation of Turing patterns. Accordingly, we confirmed that the kinetic pathways depend on competing activation ($\text{Zn}[\text{C}_5\text{H}_7\text{O}_2]_2$) (slow diffusion of an activator, coordination state), and inhibition (Fe^{3+}) (fast diffusion of an inhibitor, ionic state) refers to the theoretical model of reaction diffusion.

Researchers have found examples of Turing-like mechanisms in the distribution of species in ecosystems, such as the predator-prey model. The prey acts as an activator, seeking to reproduce and increase their numbers, while the predator acts as an inhibitor, keeping populations in check [r3]. Since the mathematical equations involved in the above model are coupled nonlinear reaction-diffusion equations, it is difficult to obtain their exact solutions. Therefore, we use the finite difference method to carry out Turing pattern simulations through MATLAB [r4]. We are now in a position to examine patterns generated from numerical simulations of the system as long as the system parameters change in the Turing space. Thus, for a given $d = \frac{d_B}{d_A} = \frac{d_{\text{Zn}}}{d_{\text{Fe}}} = \frac{1}{30}$, we expect to see the formation of patterns. Furthermore, the given parameter value according to our chemical system will fall into the Turing space, giving rise to different spatial patterns with respect to time, as shown in Fig. 3h. Turing patterns can still be obtained by changing key parameters such as diffusion coefficient difference (d) and initial conditions (Supplementary Fig. 19). The consistency between experiments and simulations verified that the formation of the architecture is closely related to Turing's theory. A movie for the numerical simulations of the Turing structures is shown in Supplementary Movie 1. Detailed formula derivation and parameter setting are shown in the Numerical Simulation section in Supplementary Information as follows:

Reaction-diffusion model for pattern formation have received increasing attention from theoretical biology and applied mathematics. The equations are of the following form:

$$\begin{cases} \frac{\partial u}{\partial t} = f(u, v) + d_u \Delta u \\ \frac{\partial v}{\partial t} = g(u, v) + d_v \Delta v \end{cases} \quad (1)$$

where u and v is the vector of concentration, the functions $f(u, v)$ and $g(u, v)$ represent the reaction kinetics (diffusion term), d_u and d_v are the diffusion coefficients of u and v , respectively. Researchers have found examples of Turing-like mechanisms in the distribution of species in ecosystems, such as the predator-prey model. In our experiment, substance A (Fe) and substance B (Zn) with large difference in diffusion coefficient correspond to inhibitor (predator)

and activator (prey), respectively. This type of model takes the form [r3]:

$$\begin{cases} \frac{du}{dt} = ru \left(1 - \frac{u}{K}\right) - \frac{muv}{u+a} \\ \frac{dv}{dt} = \frac{buv}{u+a} - cv \end{cases} \quad (2)$$

Applying the scaling to system: $rt \rightarrow t$, $\frac{u}{K} \rightarrow u$, $\frac{m}{rK}v \rightarrow v$, $\frac{a}{K} \rightarrow \alpha$, $\frac{b}{r} \rightarrow \beta$, $\frac{c}{r} \rightarrow \gamma$

Then

$$\begin{cases} \frac{du}{dt} = u(1-u) - \frac{uv}{u+\alpha} \\ \frac{dv}{dt} = \frac{\beta uv}{u+\alpha} - \gamma v \end{cases} \quad (3)$$

$$\begin{cases} u_t = d_A \Delta u + u(1-u) - \frac{uv}{u+\alpha} & x \in \Omega, t > 0 \\ v_t = d_B \Delta v + \frac{\beta uv}{u+\alpha} - \gamma v & x \in \Omega, t > 0 \\ \frac{\partial u}{\partial n} = \frac{\partial v}{\partial n} = 0 & x \in \partial\Omega, t > 0 \\ u(x, 0) = u_0 \geq 0 \quad v(x, 0) = v_0 \geq 0 & x \in \Omega \end{cases} \quad (4)$$

where u and v represent the diffusion functions of B and A in the experimental system respectively, Ω is a fixed bounded domain (substrate boundary), n is the outward unit normal vector of the boundary, Δ is Laplace operator.

The positive equilibrium point $E^*(u^*, v^*)$ can be obtained by judging the instability of equation 4.

$$u^* = \frac{\alpha\gamma}{\beta-\gamma} \quad v^* = (\alpha + u^*)(1 - u^*) \quad \beta > \gamma \quad \alpha < \frac{\beta-\gamma}{\gamma}$$

A time perturbation (t) is applied around E^* .

$$\begin{cases} \hat{u}(r, t) = u_0 e^{\lambda t} e^{i\vec{k}\vec{r}} & \hat{v}(r, t) = v_0 e^{\lambda t} e^{i\vec{k}\vec{r}} \\ |\hat{u}(r, t)| \ll u^* & |\hat{v}(r, t)| \ll v^* \end{cases} \quad (5)$$

where \vec{r} is space vector.

$$\begin{cases} g(u) = 1 - u & p(u) = \frac{u}{\alpha+u} \\ h(u) = \frac{ug(u)}{p(u)} = (1-u)(\alpha+u) \\ f_u = p(u^*)h'(u^*) & f_v = -p(u^*) \\ g_u = \frac{\gamma^2}{\beta} & g_v = -\gamma \quad d = \frac{d_B}{d_A} \quad D = \begin{pmatrix} 1 & 0 \\ 0 & d \end{pmatrix} \end{cases} \quad (6)$$

The Jacobian matrix at the positive equilibrium is shown below:

$$J(E^*) = \begin{pmatrix} p(u^*)h'(u^*) & -p(u^*) \\ \frac{\gamma^2}{\beta} & -\gamma \end{pmatrix} \quad (7)$$

We can derive from the linear system:

$$|\lambda I - J + Dk^2| = 0 \quad (8)$$

The characteristic function of equation 5 is as follows:

$$\begin{cases} \lambda^2 + \lambda[k^2(1+d) - (f_u + g_v)] + h(k^2) = 0 \\ h(k^2) = dk^4 - (df_u + g_v)k^2 + |J| \\ |J| = \gamma p(u^*) \left[\frac{\gamma}{\beta} - h'(E^*) \right] \end{cases} \quad (9)$$

The condition for a spatial mode defined by k to be unstable and thus to form a pattern in equation 9. $Re(\lambda)$ is a function of $h(k^2)$, and $h(k^2)$ is closely related with d . Therefore, $d(d = \frac{d_B}{d_A})$ determines the range of $Re(\lambda)$ and the stability of the diffusion system. Next, we use numerical simulation [r4] to verify whether the experimental system is unstable. First, we simplify equation 4 to the following form:

$$\begin{cases} f(u, v) = u(1 - u) - \frac{uv}{u+\alpha} \\ g(u, v) = \frac{\beta uv}{u+\alpha} - \gamma v \end{cases} \quad (10)$$

For the two-dimensional approximations, we use a uniform subdivision of the square by finite difference method.

$$\begin{cases} \Omega = [A, B] \times [A, B] & (x_i, y_j) = (ih + A, jh + A) \quad i, j = 0, \dots, J \\ & h = \frac{B-A}{J} \end{cases} \quad (11)$$

$\vec{U}_{i,j}^n = (U_{i,j}^n, V_{i,j}^n)^T$ denotes the two-dimensional approximation at the point (x_i, y_j, t_n) . We also carry out a uniform subdivision of the time interval $[0, T]$ with time levels $t_n = n\Delta t$, $n = 1, \dots, N$, so the time step is $\Delta t = \frac{T}{N}$. h is space step. Two-dimensional linear schemes are revealed as the following general form. For $n = 1, \dots, N$ and $i, j = 0, \dots, J$ find $\{U_{i,j}^n, V_{i,j}^n\}$

$$\begin{cases} \partial_n U_{i,j}^n = \Delta h U_{i,j}^n + \hat{f}(U_{i,j}^n, U_{i,j}^{n-1}) \\ \partial_n V_{i,j}^n = d\Delta h V_{i,j}^n + \hat{g}(U_{i,j}^n, U_{i,j}^{n-1}) \end{cases} \quad (12)$$

$U_{i,j}^0 := u_0(x_i, y_j)$ and $V_{i,j}^0 := v_0(x_i, y_j)$ can be understood as the initial position of B and A entering the solution.

Function $[X, Y, U, V, reu, rev] = fd2d_predator_prey(\alpha, \beta, \gamma, d, a, b, h, T, \Delta t)$.

We simulated the experiment with fixed parameters. $\alpha = 0.4, \beta = 2.0, \gamma = 0.6, a = 0, b = 400, h = 1, \Delta t = \frac{1}{3}, d = \frac{1}{30}$ or $d = \frac{1}{20}$

$$\begin{cases} U_{i,j}^0 = \frac{6}{35} - 2 * 10^{-7} * (X(i, j) - 0.1 * Y(i, j) - 225) * \\ \quad (X(i, j) - 0.1 * Y(i, j) - 675) \\ V_{i,j}^0 = \frac{116}{245} - 3 * 10^{-5} * (X(i, j) - 450) - 1.2 * 10^{-4} * (Y(i, j) - 150) \end{cases} \quad (13)$$

$$\begin{cases} U_{i,j}^0 = \frac{6}{35} - 2 * 10^{-7} * (X(i, j) - 180)(X(i, j) - 520) - 6 * 10^{-7} * \\ \quad (Y(i, j) - 80)(X(i, j) - 200) \\ V_{i,j}^0 = \frac{116}{245} - 3 * 10^{-5} * (X(i, j) - 350) - 6 * 10^{-5} * (Y(i, j) - 235) \end{cases} \quad (14)$$

When $T = 1, T = 5, T = 10, \dots, T = 1000$, the simulated Turing pattern can be obtained.

[r1] Z. Tan, S.F. Chen, X.S. Peng, L. Zhang, C.J. Gao. Polyamide membranes with nanoscale Turing structures for water purification. *Science* **360**, 518-521 (2018).

[r2] X.L. Zhang. et al. An efficient Turing-type $Ag_2Se-CoSe_2$ multi-interfacial oxygen-evolving electrocatalyst. *Angew. Chem. Int. Ed.* **60**, 6553-6560 (2021).

[r3] G.P. Hu, Z.S. Feng. Turing instability and pattern formation in a strongly coupled diffusive predator-prey system. *Int. J. Bifurcation and Chaos*. **30**, 2030020-1-15 (2020).

[r4] M. R. Garvie. Finite-difference schemes for reaction-diffusion equations modeling predator-prey interactions in MATLAB. *Bull. Math. Biol.* **69**, 931-956 (2007).

The related discussion has been added in the revised manuscript and the revised Supplementary Information:

Manuscript :

Theoretical simulation of Turing structure formation. Alan Turing mathematically showed that the stable state can destabilize under certain conditions and spontaneously generate spatially stationary patterns in a reaction-diffusion system, which is particularly well known in mathematical biology. Thus, the Turing pattern can be described by dimensionless reaction-diffusion equations combined with our chemical system:

$$\begin{cases} \frac{\partial u}{\partial t} = f(u, v) + d_u \Delta u \\ \frac{\partial v}{\partial t} = g(u, v) + d_v \Delta v \end{cases}$$

where u and v is the vector of concentration, the functions $f(u, v)$ and $g(u, v)$ represent the reaction kinetics (diffusion term), and d_u and d_v are the diffusion coefficients of u and v , respectively. In these systems, there are two chemical reactants that can not only interact, but also diffuse alone. Hence, the generation of the Turing pattern corresponds to the coupling of a nonlinear reaction kinetic process and a special diffusion process, which will be unstable due to the different diffusion velocities of the two factors. To demonstrate the instability caused by this diffusion, we continue to analyze the difference between the diffusion coefficients of the two substances by means of a combination of experiments and theory. As shown in Fig. 3a, it should be noted that the diffusion coefficient of $(\text{CH}_3\text{COO})^-$ was 4 times higher than the diffusion coefficient of $[\text{C}_5\text{H}_7\text{O}_2]^-$ in deuterated methanol for the 2D DOSY test due to the difference in molecular weight.

Fig. 3. Understanding the Turing structure by the reaction-diffusion equation. **a**, Two-dimensional diffusion-ordered nuclear magnetic resonance spectroscopy (2D DOSY) of a mixed solution of $\text{Fe}[\text{C}_5\text{H}_7\text{O}_2]_3$ and $\text{Zn}(\text{CH}_3\text{COO})_2$. **b**, Mean square displacement of Zn^{2+} in different solutions in Fig. 2f. The purple red and dark green curves represent the C solution ($\text{Fe}(\text{NO}_3)_3$, $\text{Zn}(\text{CH}_3\text{COO})_2$) and F solution ($\text{Fe}[\text{C}_5\text{H}_7\text{O}_2]_3$, $\text{Zn}(\text{CH}_3\text{COO})_2$), respectively. **c**, Snapshots of the dynamic process of F solution ($\text{Fe}[\text{C}_5\text{H}_7\text{O}_2]_3$, $\text{Zn}(\text{CH}_3\text{COO})_2$) at different simulation times. The orange balls are Zn^{2+} , the green balls are Fe^{3+} , the pink represents $[\text{C}_5\text{H}_7\text{O}_2]^-$ and the cyan represents CH_3COO^- . **d**, Snapshots of the dynamic process of C solution ($\text{Fe}(\text{NO}_3)_3$, $\text{Zn}(\text{CH}_3\text{COO})_2$) at different simulation times. The orange balls are Zn^{2+} , the green balls are Fe^{3+} , and purple red represents NO_3^- . **e**, SEM image of Zn: Fe=1:2.4 film. **f**, the Turing interface film (Zn: Fe=1:3.5). **g**, the Turing interface film (Zn: Fe=1:4). In the fourth row of the figure, blue represents ZnFe_2O_4 , and pink represents $\alpha\text{-Fe}_2\text{O}_3$. **h**, Patterns exhibited by prey and predator of the system at $T = 100, 600$ and 1000 , respectively. The difference in diffusion coefficient is 30 times. Scale bars: **c, d, e**, 100 nm.

To gain an atomic-level understanding of the re-coordination mechanism in homogeneous solutions, theoretical investigations, including molecular dynamics (MD) simulations, were explored. First, MD simulations are performed to study the dynamic process of Zn^{2+} in various solutions, and there is an order of magnitude difference in the diffusion coefficients of Zn^{2+} from $\text{Zn}(\text{CH}_3\text{COO})_2$ in the $\text{Fe}(\text{NO}_3)_3$ and $\text{Fe}[\text{C}_5\text{H}_7\text{O}_2]_3$ solutions (Fig. 3b). Meanwhile, it is clear that Zn^{2+} and $[\text{C}_5\text{H}_7\text{O}_2]^-$ show much higher affinity than CH_3COO^- (Fig. 3c), meaning that the dynamic process of Zn^{2+} is slow. While the movement of Zn^{2+} and Fe^{3+} was not affected by NO_3^- (Fig. 3d). The above experiments and MD results provide solid evidence that high molecular weight organic anions play a key role in homogeneous solution for formation of Turing patterns. Accordingly, we confirmed that the kinetic pathways depend on competing activation ($\text{Zn}[\text{C}_5\text{H}_7\text{O}_2]_2$) (slow diffusion of an activator, coordination state), and inhibition (Fe^{3+}) (fast diffusion of an inhibitor, ionic state) refers to the theoretical model of reaction diffusion.

In our experiment, as the concentration of $\text{Fe}[\text{C}_5\text{H}_7\text{O}_2]_3$ increased, the $\alpha\text{-Fe}_2\text{O}_3$ phase became more stripe-like, and ZnFe_2O_4 phase existed as spinel octahedral nanoparticles (Fig. 3e-g and Supplementary Figs. 16-18). However, one cannot claim the Turing pattern just because it looks like the Turing pattern, the formation of the Turing structure follows the reaction-diffusion equation mentioned earlier. Researchers have found examples of Turing-like mechanisms in the distribution of species in ecosystems, such as the predator-prey model. The prey acts as an activator, seeking to reproduce and increase their numbers, while the predator acts as an inhibitor, keeping populations in check²⁷. Since the mathematical equations involved in the above model are coupled nonlinear reaction-diffusion equations, it is difficult to obtain their exact solutions. Therefore, we use the finite difference method to carry out Turing pattern simulations through MATLAB²⁸ (detailed formula derivation and parameter setting are shown in the Numerical Simulation section in Supplementary Information). We are now in a position to examine patterns generated from numerical simulations of the system as long as the system parameters change in the Turing space. Thus, for a given $d = \frac{d_B}{d_A} = \frac{d_{Zn}}{d_{Fe}} = \frac{1}{30}$, we expect to see the formation of patterns. Furthermore, the given parameter value according to our chemical system will fall into the Turing space, giving rise to different spatial patterns with respect to time, as shown in Fig. 3h. Turing patterns can still be obtained by changing key parameters such as diffusion coefficient difference (d) and initial conditions (Supplementary Fig. 19). The consistency between experiments and simulations verified that the formation of the architecture is closely related to Turing's theory. A movie for the numerical simulations of the Turing structures is shown in Supplementary Movie 1.”

Supplementary Information:

“Numerical Simulation

Reaction-diffusion model for pattern formation have received increasing attention from theoretical biology and applied mathematics. The equations are of the following form:

$$\begin{cases} \frac{\partial u}{\partial t} = f(u, v) + d_u \Delta u \\ \frac{\partial v}{\partial t} = g(u, v) + d_v \Delta v \end{cases} \quad (1)$$

where u and v is the vector of concentration, the functions $f(u, v)$ and $g(u, v)$ represent the reaction kinetics (diffusion term), d_u and d_v are the diffusion coefficients of u and v , respectively. Researchers have found examples of Turing-like mechanisms in the distribution of species in ecosystems, such as the predator-prey model. In our experiment, substance A (Fe) and substance B (Zn) with large difference in diffusion coefficient correspond to inhibitor (predator) and activator (prey), respectively. This type of model takes the form²³:

$$\begin{cases} \frac{du}{dt} = ru \left(1 - \frac{u}{K}\right) - \frac{muv}{u+a} \\ \frac{dv}{dt} = \frac{buv}{u+a} - cv \end{cases} \quad (2)$$

Applying the scaling to system: $rt \rightarrow t$, $\frac{u}{K} \rightarrow u$, $\frac{m}{rK}v \rightarrow v$, $\frac{a}{K} \rightarrow \alpha$, $\frac{b}{r} \rightarrow \beta$, $\frac{c}{r} \rightarrow \gamma$

Then

$$\begin{cases} \frac{du}{dt} = u(1-u) - \frac{uv}{u+\alpha} \\ \frac{dv}{dt} = \frac{\beta uv}{u+\alpha} - \gamma v \end{cases} \quad (3)$$

$$\begin{cases} u_t = d_A \Delta u + u(1-u) - \frac{uv}{u+\alpha} & x \in \Omega, t > 0 \\ v_t = d_B \Delta v + \frac{\beta uv}{u+\alpha} - \gamma v & x \in \Omega, t > 0 \\ \frac{\partial u}{\partial n} = \frac{\partial v}{\partial n} = 0 & x \in \partial\Omega, t > 0 \\ u(x, 0) = u_0 \geq 0 \quad v(x, 0) = v_0 \geq 0 & x \in \Omega \end{cases} \quad (4)$$

where u and v represent the diffusion functions of B and A in the experimental system respectively, Ω is a fixed bounded domain (substrate boundary), n is the outward unit normal vector of the boundary, Δ is Laplace operator.

The positive equilibrium point $E^*(u^*, v^*)$ can be obtained by judging the instability of equation 4.

$$u^* = \frac{\alpha\gamma}{\beta-\gamma} \quad v^* = (\alpha + u^*)(1 - u^*) \quad \beta > \gamma \quad \alpha < \frac{\beta-\gamma}{\gamma}$$

A time perturbation (t) is applied around E^* .

$$\begin{cases} \hat{u}(r, t) = u_0 e^{\lambda t} e^{ik\vec{r}} & \hat{v}(r, t) = v_0 e^{\lambda t} e^{ik\vec{r}} \\ |\hat{u}(r, t)| \ll u^* & |\hat{v}(r, t)| \ll v^* \end{cases} \quad (5)$$

where \vec{r} is space vector.

$$\begin{cases} g(u) = 1 - u & p(u) = \frac{u}{\alpha+u} \\ h(u) = \frac{ug(u)}{p(u)} = (1-u)(\alpha+u) \\ f_u = p(u^*)h'(u^*) & f_v = -p(u^*) \\ g_u = \frac{\gamma^2}{\beta} & g_v = -\gamma \quad d = \frac{d_B}{d_A} \quad D = \begin{pmatrix} 1 & 0 \\ 0 & d \end{pmatrix} \end{cases} \quad (6)$$

The Jacobian matrix at the positive equilibrium is shown below:

$$J(E^*) = \begin{pmatrix} p(u^*)h'(u^*) & -p(u^*) \\ \frac{\gamma^2}{\beta} & -\gamma \end{pmatrix} \quad (7)$$

We can derive from the linear system:

$$|\lambda I - J + Dk^2| = 0 \quad (8)$$

The characteristic function of equation 5 is as follows:

$$\begin{cases} \lambda^2 + \lambda[k^2(1+d) - (f_u + g_v)] + h(k^2) = 0 \\ h(k^2) = dk^4 - (df_u + g_v)k^2 + |J| \\ |J| = \gamma p(u^*) \left[\frac{\gamma}{\beta} - h'(E^*) \right] \end{cases} \quad (9)$$

The condition for a spatial mode defined by k to be unstable and thus to form a pattern in equation 9. $Re(\lambda)$ is a function of $h(k^2)$, and $h(k^2)$ is closely related with d . Therefore, $d(d = \frac{d_B}{d_A})$ determines the range of $Re(\lambda)$ and the stability of the diffusion system. Next, we use numerical simulation²⁴ to verify whether the experimental system is unstable. First, we simplify equation 4 to the following form:

$$\begin{cases} f(u, v) = u(1 - u) - \frac{uv}{u+\alpha} \\ g(u, v) = \frac{\beta uv}{u+\alpha} - \gamma v \end{cases} \quad (10)$$

For the two-dimensional approximations, we use a uniform subdivision of the square by finite difference method.

$$\begin{cases} \Omega = [A, B] \times [A, B] & (x_i, y_j) = (ih + A, jh + A) \quad i, j = 0, \dots, J \\ h = \frac{B-A}{J} \end{cases} \quad (11)$$

$\vec{U}_{i,j}^n = (U_{i,j}^n, V_{i,j}^n)^T$ denotes the two-dimensional approximation at the point (x_i, y_j, t_n) . We also carry out a uniform subdivision of the time interval $[0, T]$ with time levels $t_n = n\Delta t$, $n = 1, \dots, N$, so the time step is $\Delta t = \frac{T}{N}$. h is space step. Two-dimensional linear schemes are revealed as the following general form. For $n = 1, \dots, N$ and $i, j = 0, \dots, J$ find $\{U_{i,j}^n, V_{i,j}^n\}$

$$\begin{cases} \partial_n U_{i,j}^n = \Delta h U_{i,j}^n + \hat{f}(U_{i,j}^n, U_{i,j}^{n-1}) \\ \partial_n V_{i,j}^n = d\Delta h V_{i,j}^n + \hat{g}(U_{i,j}^n, U_{i,j}^{n-1}) \end{cases} \quad (12)$$

$U_{i,j}^0 := u_0(x_i, y_j)$ and $V_{i,j}^0 := v_0(x_i, y_j)$ can be understood as the initial position of B and A entering the solution.

Function $[X, Y, U, V, reu, rev] = fd2d_predator_prey(\alpha, \beta, \gamma, d, a, b, h, T, \Delta t)$.

We simulated the experiment with fixed parameters. $\alpha = 0.4, \beta = 2.0, \gamma = 0.6, a = 0, b = 400, h = 1, \Delta t = \frac{1}{3}, d = \frac{1}{30}$ or $d = \frac{1}{20}$

$$\begin{cases} U_{i,j}^0 = \frac{6}{35} - 2 * 10^{-7} * (X(i, j) - 0.1 * Y(i, j) - 225) * \\ \quad (X(i, j) - 0.1 * Y(i, j) - 675) \\ V_{i,j}^0 = \frac{116}{245} - 3 * 10^{-5} * (X(i, j) - 450) - 1.2 * 10^{-4} * (Y(i, j) - 150) \end{cases} \quad (13)$$

$$\begin{cases} U_{i,j}^0 = \frac{6}{35} - 2 * 10^{-7} * (X(i, j) - 180)(X(i, j) - 520) - 6 * 10^{-7} * \\ \quad (Y(i, j) - 80)(X(i, j) - 200) \\ V_{i,j}^0 = \frac{116}{245} - 3 * 10^{-5} * (X(i, j) - 350) - 6 * 10^{-5} * (Y(i, j) - 235) \end{cases} \quad (14)$$

When $T = 1, T = 5, T = 10, \dots, T = 1000$, the simulated Turing pattern can be obtained.

Supplementary Fig. 19. Numerical simulation. a, Patterns exhibited by prey and predator of the

system at $T = 1, 100, 600$ and 1000 , respectively. $d = \frac{1}{30}$, and the initial data is in line with the equation 13. **b**, Patterns exhibited by prey and predator of the system at $T = 1, 100, 600$ and 1000 , respectively. $d = \frac{1}{20}$, and the initial data is in line with the equation 14.”

In addition to this, the following statements are overstatements because there is no enough evidence to support their proposal:

"we ... proposed a new concept of a Turing interface at the atomic level"

"a new concept of a Turing interface at the atomic level was presented for the first time,"

Response: Thanks for the reviewer’s valuable comments. We have modified the above statement as follows:

“which was named Turing interface at the atomic level”

“the Turing interface at the atomic level was revealed.”

From the viewpoint of readability, the manuscript is not well written. They cite SI too often, preventing readers from understanding the essence of the paper. The main text itself should powerfully state the author's views.

For these reasons, I cannot recommend this manuscript for publication in Nature Communications.

Response: We thank the reviewer to point this out. According to the suggestion, we have moved the evidence from the Supporting Information to the main text and adjusted some figures in the Supporting Information, which can strongly state our views.

(1). More information about interfacial strain was added to the main text, as shown in Figure 5i and j

Fig. 5. Atom-level Turing interface. **a**, The oxygen-bridged coordination network. **b**, HAADF-STEM image of the Turing interface (Zn: Fe=1:3) and GPA analysis of different plane strains. The interface is marked with a white dashed line in the strain graph, and the curve directly below the graph corresponds to the strain value across the interface. **c**, TEM image of the conventional dual-phase interface (Zn: Fe=1:3) and GPA analysis of different plane strains. The interface is marked with a purple dashed line in the strain graph, and the curve directly below the graph corresponds to the strain value across the interface. **d**, Crystallographic models of the interface before the phase transition and after the phase transition. **e**, Raman spectra for the β - Fe_2O_3 powder, α - Fe_2O_3 powder and the unannealed film (β - Fe_2O_3). **f**, HAADF-STEM images for the unannealed and calcined films for the only iron source ($\text{Fe}[\text{C}_5\text{H}_7\text{O}_2]_3$). **g**, Image of a larger view in the dotted box (Supplementary Fig. 28a) and the semi-coherent interface. **h**, Image of a larger view in the dotted box (Supplementary Fig. 28b). **i**, GPA analysis of different plane strains for **g**. The interface is marked with a white dashed line in the strain graph, and the curve directly below the graph corresponds to the strain value across the interface. **j**, GPA analysis of different plane strains for **h**. The interface has been marked with a white dashed line in the strain graph, and the curve directly below the graph corresponds to the strain value across the interface. **k**, the semi-coherent interface. Scale bars: **b**, 5 nm; **c**, 10 nm; **f**, 5 nm.

(2). A description of the thickness of the film was added to the main text as follows:

“In addition, the optimized film thicknesses were kept at 300 nm, 300 nm and 3.2 μm for the

Turing interface films, non-Turing dual-phase interface films and conventional dual-phase interface films (Supplementary Fig. 35b).”

(3). A description of the surface roughness of films and the influence of fabrication process was added to the main text as follows:

“To exclude the influence of the surface structure of the film, we conducted a surface roughness measurement. According to the AFM micrographs (2D and 3D) in Supplementary Fig. 46 and 47, there is little difference between the Turing structure film and the non-Turing structure film, and the surface roughness is distributed at 5-8 nm. Therefore, we can be sure that the same preparation process has little effect on the surface roughness of the film. Moreover, there is no strong dependence between its performance and surface roughness. We confirm that the enhancement in the efficiency of the thin film is attributed to the formation of the bulk Turing interface. In order to further verify that the formation of the Turing structure is determined by the intrinsic property of the solution rather than an artifact that can appear due to the fabrication process, we used the drop coating and spin coating technique to prepare high-performance Turing structure films (see detailed information in Supplementary Figs. 48 and 49).”

(4). The KPFM characterization of samples was combined in the revised Supplementary Information.

Supplementary Fig. 44. KPFM and C-AFM. a, Topographic AFM image of the Turing interface film (Zn: Fe=1:3) for a V_{CPD} . b, a V_{CPD} image in the dark for the Turing interface film (Zn: Fe=1:3). c, An AFM image of α - Fe_2O_3 film. d, a V_{CPD} image of α - Fe_2O_3 film in the dark. e, a V_{CPD} image of α - Fe_2O_3 film under light irradiation. f, Topographic AFM image of the Turing interface film (Zn: Fe=1:3) for current measurement. g, A current image of the Turing interface film (Zn: Fe=1:3) in the dark (1 V bias). h, A current image of the Turing interface film (Zn: Fe=1:3) under 405 nm light irradiation (1 V bias). i, Current images of the Turing interface film (Zn: Fe=1:3) overlaid with (3D) in the dark (1 V bias).

(5). The characterizations of $MgFe_2O_4$ and $CuFe_2O_4$ films and were combined in the revised Supplementary Information.

Supplementary Fig. 54. SEM images and elemental mappings of Turing interface film. a, CuFe₂O₄. **b,** Turing interface film (Cu: Fe=1:3), the illustration shows the pseudo-color mode. **c,** MgFe₂O₄. **d,** Turing interface film (Mg: Fe=1:3), the illustration shows the pseudo-color mode. **e,** Crystallographic images of MgFe₂O₄. **f,** J-V curves of photoelectrode films of MgFe₂O₄. **g,** Crystallographic images of CuFe₂O₄. **h,** J-V curves of photoelectrode films of CuFe₂O₄.

Reviewer #2 (Remarks to the Author):

This manuscript (NCOMMS-21-39616) describes the synthesis of near-zero strain semi-coherent Turing interface spinel ferrites films with various Zn: Fe, Cu: Fe and Mg: Fe ratios with enhanced photoelectrochemical activities. This work provides the details of the synthesis of Turing interface spinel ferrites film and its advantages and related characterization for solar water splitting and photoelectrochemical phenol oxidation for the first time. This work elaborates a new insight and clear understanding of using Turing interface spinel ferrites films for energy-related applications and related areas. Hence, considering the detailed systematic studies carried out in this work, well supported by experiments and theoretical studies, this manuscript may be considered for publication in Nature Comm after the appropriate revisions as mentioned below. The authors should include and address the following points in the revised manuscript clearly.

Response: We thank the reviewer for the very positive assessment of our work. The comments lead to further improve the quality of our work. According to the comments, we have modified our manuscript discussion and corresponding responses.

1. How can one confirm that the Turing structure is formed? Is there any direct study/simulation etc. to support the claim.

Response: We thank the reviewer for the insightful comments. The formation of Turing structure depends on whether there are substances with order of magnitude difference in diffusion coefficient in homogeneous solution, and the instability of system can be controlled [r1-r3]. In our homogeneous solution, the formation of Turing structure is determined by intrinsic property of the solution rather than an artifact that can appear due to the fabrication process as shown in Supplementary Fig. 48 (Added experimental contents). We used the drop coating and spin coating technique to prepare high-performance Turing structure films. Furthermore, we conducted the experiments and simulations, including two-dimensional diffusion-ordered nuclear magnetic resonance spectroscopy (2D DOSY) (Fig. 3a), molecular dynamics (MD) simulation (Fig. 3b-d) and numerical simulation (Fig. 3h and Supplementary Fig. 19). Meanwhile, samples prepared from $\text{Fe}(\text{NO}_3)_3$ and FeCl_3 were provided as non-Turing structures. A movie for the numerical simulations of the Turing structures is shown in Supplementary Movie 1. A movie of the shape evolution of droplets is provided as a newly added Supplementary Movie 2. We will analyze these results in detail below.

Supplementary Fig. 48. Different fabrication strategies were used to verify the homogeneous solution characteristics of Turing structure and non-Turing structure. Drop coating: a, The shape evolution of droplets (Zn: Fe=1:3, $\text{Fe}[\text{C}_5\text{H}_7\text{O}_2]_3$). b, SEM image and elemental mapping of Turing structure film for above solution. c, The shape evolution of droplets (Zn: Fe=1:3, $\text{Fe}(\text{NO}_3)_3$). d, SEM image and elemental mapping of non-Turing dual-phase interface film for above solution. e, The shape evolution of droplets (Zn: Fe=1:3, FeCl_3). f, SEM image and elemental mapping of non-Turing dual-phase interface films for above solution. g, XRD patterns for Turing structure film and non-Turing structure film. h, Current density-potential profiles of Turing structure film and non-Turing structure film. **Spin coating:** i, The shape evolution of droplets (Zn: Fe=1:3, $\text{Fe}[\text{C}_5\text{H}_7\text{O}_2]_3$). j, SEM image and elemental mapping of Turing structure film for above solution. k, The shape evolution of droplets (Zn: Fe=1:3, $\text{Fe}(\text{NO}_3)_3$). l, SEM image and elemental mapping of non-Turing dual-phase interface films for above solution. m, The shape evolution of droplets (Zn: Fe=1:3, FeCl_3). n, SEM image and elemental mapping of non-Turing dual-phase interface films for above solution. g, XRD patterns for Turing structure film and non-Turing structure film. h, Current density-potential profiles of Turing structure film and non-Turing structure film. Scale bars: b, d, f, j, l, n, 200 nm.

Fig. 3. Understanding the Turing structure by the reaction-diffusion equation. **a**, Two-dimensional diffusion-ordered nuclear magnetic resonance spectroscopy (2D DOSY) of a mixed solution of $\text{Fe}[\text{C}_5\text{H}_7\text{O}_2]_3$ and $\text{Zn}(\text{CH}_3\text{COO})_2$. **b**, Mean square displacement of Zn^{2+} in different solutions in Fig. 2f. The purple red and dark green curves represent the C solution ($\text{Fe}(\text{NO}_3)_3$, $\text{Zn}(\text{CH}_3\text{COO})_2$) and F solution ($\text{Fe}[\text{C}_5\text{H}_7\text{O}_2]_3$, $\text{Zn}(\text{CH}_3\text{COO})_2$), respectively. **c**, Snapshots of the dynamic process of F solution ($\text{Fe}[\text{C}_5\text{H}_7\text{O}_2]_3$, $\text{Zn}(\text{CH}_3\text{COO})_2$) at different simulation times. The orange balls are Zn^{2+} , the green balls are Fe^{3+} , the pink represents $[\text{C}_5\text{H}_7\text{O}_2]^-$ and the cyan represents CH_3COO^- . **d**, Snapshots of the dynamic process of C solution ($\text{Fe}(\text{NO}_3)_3$, $\text{Zn}(\text{CH}_3\text{COO})_2$) at different simulation times. The orange balls are Zn^{2+} , the green balls are Fe^{3+} , and purple red represents NO_3^- . **e**, SEM image of Zn: Fe=1:2.4 film. **f**, the Turing interface film (Zn: Fe=1:3.5). **g**, the Turing interface film (Zn: Fe=1:4). In the fourth row of the figure, blue represents ZnFe_2O_4 , and pink represents $\alpha\text{-Fe}_2\text{O}_3$. **h**, Patterns exhibited by prey and predator of the system at $T = 100, 600$ and 1000 , respectively. The difference in diffusion coefficient is 30 times.

Scale bars: **c, d, e**, 100 nm.

Supplementary Fig. 19. Numerical simulation. **a**, Patterns exhibited by prey and predator of the system at $T = 1, 100, 600$ and 1000 , respectively. $d = \frac{1}{30}$, and the initial data is in line with the equation 13. **b**, Patterns exhibited by prey and predator of the system at $T = 1, 100, 600$ and 1000 , respectively. $d = \frac{1}{20}$, and the initial data is in line with the equation 14.

Turing structures typically emerge in reaction-diffusion processes far from thermodynamic equilibrium, involving at least two chemicals with different diffusion coefficients (inhibitors and activators) [r_1, r_2]. Alan Turing mathematically showed that the stable state can destabilize under certain conditions and spontaneously generate spatially stationary patterns in a reaction-diffusion system, which is particularly well known in mathematical biology. Thus, the Turing pattern can be described by dimensionless reaction-diffusion equations combined with our chemical system:

$$\begin{cases} \frac{\partial u}{\partial t} = f(u, v) + d_u \Delta u \\ \frac{\partial v}{\partial t} = g(u, v) + d_v \Delta v \end{cases}$$

where u and v is the vector of concentration, the functions $f(u, v)$ and $g(u, v)$ represent the reaction kinetics (diffusion term), and d_u and d_v are the diffusion coefficients of u and v , respectively. The generation of the Turing pattern corresponds to the coupling of a nonlinear reaction kinetic process and a special diffusion process, which will be unstable due to the different diffusion velocities of the two factors.

First, we verified the activators and inhibitors with big difference in diffusion coefficient in homogeneous solution by the 2D DOSY and MD simulation. As shown in Fig. 3a, it should be noted that the diffusion coefficient of $(\text{CH}_3\text{COO})^-$ was 4 times higher than the diffusion coefficient of $[\text{C}_5\text{H}_7\text{O}_2]^-$ in deuterated methanol for the 2D DOSY test due to the difference in molecular weight. Next, to gain an atomic-level understanding of the re-coordination mechanism in homogeneous solutions, MD simulations were explored. There is an order of magnitude difference in the diffusion coefficients of Zn^{2+} from $\text{Zn}(\text{CH}_3\text{COO})_2$ in the $\text{Fe}(\text{NO}_3)_3$ and $\text{Fe}[\text{C}_5\text{H}_7\text{O}_2]_3$ solutions (Fig. 3b). Meanwhile, it is clear that Zn^{2+} and $[\text{C}_5\text{H}_7\text{O}_2]^-$ show much higher affinity than CH_3COO^- (Fig. 3c), meaning that the dynamic process of Zn^{2+} is slow. While the movement of Zn^{2+} and Fe^{3+} was not affected by NO_3^- (Fig. 3d). The above experiments and MD simulations

results provide solid evidence that high molecular weight organic anions play a key role in homogeneous solution for formation of Turing patterns. Accordingly, we confirmed that the kinetic pathways depend on competing activation ($\text{Zn}[\text{C}_5\text{H}_7\text{O}_2]_2$) (slow diffusion of an activator, coordination state), and inhibition (Fe^{3+}) (fast diffusion of an inhibitor, ionic state) refers to the theoretical model of reaction diffusion.

Researchers have found examples of Turing-like mechanisms in the distribution of species in ecosystems, such as the predator-prey model. The prey acts as an activator, seeking to reproduce and increase their numbers, while the predator acts as an inhibitor, keeping populations in check [r4]. Since the mathematical equations involved in the above model are coupled nonlinear reaction-diffusion equations, it is difficult to obtain their exact solutions. Therefore, we use the finite difference method to carry out Turing pattern simulations through MATLAB [r5]. We are now in a position to examine patterns generated from numerical simulations of the system as long as the system parameters change in the Turing space. Thus, for a given $d = \frac{d_B}{d_A} = \frac{d_{Zn}}{d_{Fe}} = \frac{1}{30}$, we expect to see the formation of patterns. Furthermore, the given parameter value according to our chemical system will fall into the Turing space, giving rise to different spatial patterns with respect to time, as shown in Fig. 3h. Turing patterns can still be obtained by changing key parameters such as diffusion coefficient difference (d) and initial conditions (Supplementary Fig. 19). The consistency between experiments and simulations verified that the formation of the architecture is closely related to Turing's theory. Detailed formula derivation and parameter setting are shown in the Numerical Simulation section in Supplementary Information as follows:

Reaction-diffusion model for pattern formation have received increasing attention from theoretical biology and applied mathematics. The equations are of the following form:

$$\begin{cases} \frac{\partial u}{\partial t} = f(u, v) + d_u \Delta u \\ \frac{\partial v}{\partial t} = g(u, v) + d_v \Delta v \end{cases} \quad (1)$$

where u and v is the vector of concentration, the functions $f(u, v)$ and $g(u, v)$ represent the reaction kinetics (diffusion term), d_u and d_v are the diffusion coefficients of u and v , respectively. Researchers have found examples of Turing-like mechanisms in the distribution of species in ecosystems, such as the predator-prey model. In our experiment, substance A (Fe) and substance B (Zn) with large difference in diffusion coefficient correspond to inhibitor (predator) and activator (prey), respectively. This type of model takes the form [r3]:

$$\begin{cases} \frac{du}{dt} = ru \left(1 - \frac{u}{K}\right) - \frac{muv}{u+a} \\ \frac{dv}{dt} = \frac{buv}{u+a} - cv \end{cases} \quad (2)$$

Applying the scaling to system: $rt \rightarrow t$, $\frac{u}{K} \rightarrow u$, $\frac{m}{rK} v \rightarrow v$, $\frac{a}{K} \rightarrow \alpha$, $\frac{b}{r} \rightarrow \beta$, $\frac{c}{r} \rightarrow \gamma$

Then

$$\begin{cases} \frac{du}{dt} = u(1-u) - \frac{uv}{u+\alpha} \\ \frac{dv}{dt} = \frac{\beta uv}{u+\alpha} - \gamma v \end{cases} \quad (3)$$

$$\begin{cases} u_t = d_A \Delta u + u(1-u) - \frac{uv}{u+\alpha} & x \in \Omega, t > 0 \\ v_t = d_B \Delta v + \frac{\beta uv}{u+\alpha} - \gamma v & x \in \Omega, t > 0 \\ \frac{\partial u}{\partial n} = \frac{\partial v}{\partial n} = 0 & x \in \partial\Omega, t > 0 \\ u(x, 0) = u_0 \geq 0 \quad v(x, 0) = v_0 \geq 0 & x \in \Omega \end{cases} \quad (4)$$

where u and v represent the diffusion functions of B and A in the experimental system respectively, Ω is a fixed bounded domain (substrate boundary), n is the outward unit normal vector of the boundary, Δ is Laplace operator.

The positive equilibrium point $E^*(u^*, v^*)$ can be obtained by judging the instability of equation 4.

$$u^* = \frac{\alpha\gamma}{\beta-\gamma} \quad v^* = (\alpha + u^*)(1 - u^*) \quad \beta > \gamma \quad \alpha < \frac{\beta-\gamma}{\gamma}$$

A time perturbation (t) is applied around E^* .

$$\begin{cases} \hat{u}(r, t) = u_0 e^{\lambda t} e^{ik\vec{r}} & \hat{v}(r, t) = v_0 e^{\lambda t} e^{ik\vec{r}} \\ |\hat{u}(r, t)| \ll u^* & |\hat{v}(r, t)| \ll v^* \end{cases} \quad (5)$$

where \vec{r} is space vector.

$$\begin{cases} g(u) = 1 - u & p(u) = \frac{u}{\alpha+u} \\ h(u) = \frac{ug(u)}{p(u)} = (1-u)(\alpha+u) \\ f_u = p(u^*)h'(u^*) & f_v = -p(u^*) \\ g_u = \frac{\gamma^2}{\beta} & g_v = -\gamma \quad d = \frac{d_B}{d_A} \quad D = \begin{pmatrix} 1 & 0 \\ 0 & d \end{pmatrix} \end{cases} \quad (6)$$

The Jacobian matrix at the positive equilibrium is shown below:

$$J(E^*) = \begin{pmatrix} p(u^*)h'(u^*) & -p(u^*) \\ \frac{\gamma^2}{\beta} & -\gamma \end{pmatrix} \quad (7)$$

We can derive from the linear system:

$$|\lambda I - J + Dk^2| = 0 \quad (8)$$

The characteristic function of equation 5 is as follows:

$$\begin{cases} \lambda^2 + \lambda[k^2(1+d) - (f_u + g_v)] + h(k^2) = 0 \\ h(k^2) = dk^4 - (df_u + g_v)k^2 + |J| \\ |J| = \gamma p(u^*) \left[\frac{\gamma}{\beta} - h'(E^*) \right] \end{cases} \quad (9)$$

The condition for a spatial mode defined by k to be unstable and thus to form a pattern in equation 9. $Re(\lambda)$ is a function of $h(k^2)$, and $h(k^2)$ is closely related with d . Therefore, $d(d = \frac{d_B}{d_A})$ determines the range of $Re(\lambda)$ and the stability of the diffusion system. Next, we use

numerical simulation [r4] to verify whether the experimental system is unstable. First, we simplify equation 4 to the following form:

$$\begin{cases} f(u, v) = u(1 - u) - \frac{uv}{u+\alpha} \\ g(u, v) = \frac{\beta uv}{u+\alpha} - \gamma v \end{cases} \quad (10)$$

For the two-dimensional approximations, we use a uniform subdivision of the square by finite difference method.

$$\begin{cases} \Omega = [A, B] \times [A, B] & (x_i, y_j) = (ih + A, jh + A) \quad i, j = 0, \dots, J \\ h = \frac{B-A}{J} \end{cases} \quad (11)$$

$\vec{U}_{i,j}^n = (U_{i,j}^n, V_{i,j}^n)^T$ denotes the two-dimensional approximation at the point (x_i, y_j, t_n) . We also carry out a uniform subdivision of the time interval $[0, T]$ with time levels $t_n = n\Delta t$, $n = 1, \dots, N$, so the time step is $\Delta t = \frac{T}{N}$. h is space step. Two-dimensional linear schemes are revealed as the following general form. For $n = 1, \dots, N$ and $i, j = 0, \dots, J$ find $\{U_{i,j}^n, V_{i,j}^n\}$

$$\begin{cases} \partial_n U_{i,j}^n = \Delta h U_{i,j}^n + \hat{f}(U_{i,j}^n, U_{i,j}^{n-1}) \\ \partial_n V_{i,j}^n = d\Delta h V_{i,j}^n + \hat{g}(U_{i,j}^n, U_{i,j}^{n-1}) \end{cases} \quad (12)$$

$U_{i,j}^0 := u_0(x_i, y_j)$ and $V_{i,j}^0 := v_0(x_i, y_j)$ can be understood as the initial position of B and A entering the solution.

Function $[X, Y, U, V, \text{reu}, \text{rev}] = fd2d_predator_prey(\alpha, \beta, \gamma, d, a, b, h, T, \Delta t)$.

We simulated the experiment with fixed parameters. $\alpha = 0.4, \beta = 2.0, \gamma = 0.6, a = 0, b = 400, h = 1, \Delta t = \frac{1}{3}, d = \frac{1}{30}$ or $d = \frac{1}{20}$

$$\begin{cases} U_{i,j}^0 = \frac{6}{35} - 2 * 10^{-7} * (X(i, j) - 0.1 * Y(i, j) - 225) * \\ \quad (X(i, j) - 0.1 * Y(i, j) - 675) \\ V_{i,j}^0 = \frac{116}{245} - 3 * 10^{-5} * (X(i, j) - 450) - 1.2 * 10^{-4} * (Y(i, j) - 150) \end{cases} \quad (13)$$

$$\begin{cases} U_{i,j}^0 = \frac{6}{35} - 2 * 10^{-7} * (X(i, j) - 180)(X(i, j) - 520) - 6 * 10^{-7} * \\ \quad (Y(i, j) - 80)(X(i, j) - 200) \\ V_{i,j}^0 = \frac{116}{245} - 3 * 10^{-5} * (X(i, j) - 350) - 6 * 10^{-5} * (Y(i, j) - 235) \end{cases} \quad (14)$$

When $T = 1, T = 5, T = 10, \dots, T = 1000$, the simulated Turing pattern can be obtained.

[r1] Z. Tan, S.F. Chen, X.S. Peng, L. Zhang, C.J. Gao. Polyamide membranes with nanoscale Turing structures for water purification. *Science* **360**, 518-521 (2018).

[r2] X.L. Zhang. et al. An efficient Turing-type $\text{Ag}_2\text{Se-CoSe}_2$ multi-interfacial oxygen-evolving electrocatalyst. *Angew. Chem. Int. Ed.* **60**, 6553-6560 (2021).

[r3] Y. Fuseya, H. Katsuno, K. Behnia, A. Kapitulnik. Nanoscale Turing patterns in a bismuth monolayer. *Nat. Phys.* **17**, 1031-1036 (2021).

[r4] G.P. Hu, Z.S. Feng. Turing instability and pattern formation in a strongly coupled diffusive predator-prey system. *Int. J. Bifurcation and Chaos.* **30**, 2030020-1-15 (2020).

[r5] M. R. Garvie. Finite-difference schemes for reaction-diffusion equations modeling predator-prey interactions in MATLAB. *Bull. Math. Biol.* **69**, 931-956 (2007).

The related discussion has been added in the revised manuscript and the revised Supplementary Information:
 Manuscript :

“Theoretical simulation of Turing structure formation. Alan Turing mathematically showed that the stable state can destabilize under certain conditions and spontaneously generate spatially stationary patterns in a reaction-diffusion system, which is particularly well known in mathematical biology. Thus, the Turing pattern can be described by dimensionless reaction-diffusion equations combined with our chemical system:

$$\begin{cases} \frac{\partial u}{\partial t} = f(u, v) + d_u \Delta u \\ \frac{\partial v}{\partial t} = g(u, v) + d_v \Delta v \end{cases}$$

where u and v is the vector of concentration, the functions $f(u, v)$ and $g(u, v)$ represent the reaction kinetics (diffusion term), and d_u and d_v are the diffusion coefficients of u and v , respectively. In these systems, there are two chemical reactants that can not only interact, but also diffuse alone. Hence, the generation of the Turing pattern corresponds to the coupling of a nonlinear reaction kinetic process and a special diffusion process, which will be unstable due to the different diffusion velocities of the two factors. To demonstrate the instability caused by this diffusion, we continue to analyze the difference between the diffusion coefficients of the two substances by means of a combination of experiments and theory. As shown in Fig. 3a, it should be noted that the diffusion coefficient of $(\text{CH}_3\text{COO})^-$ was 4 times higher than the diffusion coefficient of $[\text{C}_5\text{H}_7\text{O}_2]^-$ in deuterated methanol for the 2D DOSY test due to the difference in molecular weight.

Fig. 3. Understanding the Turing structure by the reaction-diffusion equation. a, Two-dimensional diffusion-ordered nuclear magnetic resonance spectroscopy (2D DOSY) of a mixed solution of $\text{Fe}[\text{C}_5\text{H}_7\text{O}_2]_3$ and $\text{Zn}(\text{CH}_3\text{COO})_2$. **b,** Mean square displacement of Zn^{2+} in

different solutions in Fig. 2f. The purple red and dark green curves represent the C solution ($\text{Fe}(\text{NO}_3)_3$, $\text{Zn}(\text{CH}_3\text{COO})_2$) and F solution ($\text{Fe}[\text{C}_5\text{H}_7\text{O}_2]_3$, $\text{Zn}(\text{CH}_3\text{COO})_2$), respectively. **c**, Snapshots of the dynamic process of F solution ($\text{Fe}[\text{C}_5\text{H}_7\text{O}_2]_3$, $\text{Zn}(\text{CH}_3\text{COO})_2$) at different simulation times. The orange balls are Zn^{2+} , the green balls are Fe^{3+} , the pink represents $[\text{C}_5\text{H}_7\text{O}_2]^-$ and the cyan represents CH_3COO^- . **d**, Snapshots of the dynamic process of C solution ($\text{Fe}(\text{NO}_3)_3$, $\text{Zn}(\text{CH}_3\text{COO})_2$) at different simulation times. The orange balls are Zn^{2+} , the green balls are Fe^{3+} , and purple red represents NO_3^- . **e**, SEM image of Zn: Fe=1:2.4 film. **f**, the Turing interface film (Zn: Fe=1:3.5). **g**, the Turing interface film (Zn: Fe=1:4). In the fourth row of the figure, blue represents ZnFe_2O_4 , and pink represents $\alpha\text{-Fe}_2\text{O}_3$. **h**, Patterns exhibited by prey and predator of the system at $T = 100, 600$ and 1000 , respectively. The difference in diffusion coefficient is 30 times. Scale bars: **c, d, e**, 100 nm.

To gain an atomic-level understanding of the re-coordination mechanism in homogeneous solutions, theoretical investigations, including molecular dynamics (MD) simulations, were explored. First, MD simulations are performed to study the dynamic process of Zn^{2+} in various solutions, and there is an order of magnitude difference in the diffusion coefficients of Zn^{2+} from $\text{Zn}(\text{CH}_3\text{COO})_2$ in the $\text{Fe}(\text{NO}_3)_3$ and $\text{Fe}[\text{C}_5\text{H}_7\text{O}_2]_3$ solutions (Fig. 3b). Meanwhile, it is clear that Zn^{2+} and $[\text{C}_5\text{H}_7\text{O}_2]^-$ show much higher affinity than CH_3COO^- (Fig. 3c), meaning that the dynamic process of Zn^{2+} is slow. While the movement of Zn^{2+} and Fe^{3+} was not affected by NO_3^- (Fig. 3d). The above experiments and MD results provide solid evidence that high molecular weight organic anions play a key role in homogeneous solution for formation of Turing patterns. Accordingly, we confirmed that the kinetic pathways depend on competing activation ($\text{Zn}[\text{C}_5\text{H}_7\text{O}_2]_2$) (slow diffusion of an activator, coordination state), and inhibition (Fe^{3+}) (fast diffusion of an inhibitor, ionic state) refers to the theoretical model of reaction diffusion.

In our experiment, as the concentration of $\text{Fe}[\text{C}_5\text{H}_7\text{O}_2]_3$ increased, the $\alpha\text{-Fe}_2\text{O}_3$ phase became more stripe-like, and ZnFe_2O_4 phase existed as spinel octahedral nanoparticles (Fig. 3e-g and Supplementary Figs. 16-18). However, one cannot claim the Turing pattern just because it looks like the Turing pattern, the formation of the Turing structure follows the reaction-diffusion equation mentioned earlier. Researchers have found examples of Turing-like mechanisms in the distribution of species in ecosystems, such as the predator-prey model. The prey acts as an activator, seeking to reproduce and increase their numbers, while the predator acts as an inhibitor, keeping populations in check²⁷. Since the mathematical equations involved in the above model are coupled nonlinear reaction-diffusion equations, it is difficult to obtain their exact solutions. Therefore, we use the finite difference method to carry out Turing pattern simulations through MATLAB²⁸ (detailed formula derivation and parameter setting are shown in the Numerical Simulation section in Supplementary Information). We are now in a position to examine patterns generated from numerical simulations of the system as long as the system parameters change in the Turing space. Thus, for a given $d = \frac{d_B}{d_A} = \frac{d_{Zn}}{d_{Fe}} = \frac{1}{30}$, we expect to see the formation of patterns.

Furthermore, the given parameter value according to our chemical system will fall into the Turing

space, giving rise to different spatial patterns with respect to time, as shown in Fig. 3h. Turing patterns can still be obtained by changing key parameters such as diffusion coefficient difference (d) and initial conditions (Supplementary Fig. 19). The consistency between experiments and simulations verified that the formation of the architecture is closely related to Turing's theory. A movie for the numerical simulations of the Turing structures is shown in Supplementary Movie 1."

Supplementary Information:

"Numerical Simulation

Reaction-diffusion model for pattern formation have received increasing attention from theoretical biology and applied mathematics. The equations are of the following form:

$$\begin{cases} \frac{\partial u}{\partial t} = f(u, v) + d_u \Delta u \\ \frac{\partial v}{\partial t} = g(u, v) + d_v \Delta v \end{cases} \quad (1)$$

where u and v is the vector of concentration, the functions $f(u, v)$ and $g(u, v)$ represent the reaction kinetics (diffusion term), d_u and d_v are the diffusion coefficients of u and v , respectively. Researchers have found examples of Turing-like mechanisms in the distribution of species in ecosystems, such as the predator-prey model. In our experiment, substance A (Fe) and substance B (Zn) with large difference in diffusion coefficient correspond to inhibitor (predator) and activator (prey), respectively. This type of model takes the form²³:

$$\begin{cases} \frac{du}{dt} = ru \left(1 - \frac{u}{K}\right) - \frac{muv}{u+a} \\ \frac{dv}{dt} = \frac{buv}{u+a} - cv \end{cases} \quad (2)$$

Applying the scaling to system: $rt \rightarrow t$, $\frac{u}{K} \rightarrow u$, $\frac{m}{rK} v \rightarrow v$, $\frac{a}{K} \rightarrow \alpha$, $\frac{b}{r} \rightarrow \beta$, $\frac{c}{r} \rightarrow \gamma$

Then

$$\begin{cases} \frac{du}{dt} = u(1 - u) - \frac{uv}{u+\alpha} \\ \frac{dv}{dt} = \frac{\beta uv}{u+\alpha} - \gamma v \end{cases} \quad (3)$$

$$\begin{cases} u_t = d_A \Delta u + u(1 - u) - \frac{uv}{u+\alpha} & x \in \Omega, t > 0 \\ v_t = d_B \Delta v + \frac{\beta uv}{u+\alpha} - \gamma v & x \in \Omega, t > 0 \\ \frac{\partial u}{\partial n} = \frac{\partial v}{\partial n} = 0 & x \in \partial\Omega, t > 0 \\ u(x, 0) = u_0 \geq 0 \quad v(x, 0) = v_0 \geq 0 & x \in \Omega \end{cases} \quad (4)$$

where u and v represent the diffusion functions of B and A in the experimental system respectively, Ω is a fixed bounded domain (substrate boundary), n is the outward unit normal vector of the boundary, Δ is Laplace operator.

The positive equilibrium point $E^*(u^*, v^*)$ can be obtained by judging the instability of equation 4.

$$u^* = \frac{\alpha\gamma}{\beta-\gamma} \quad v^* = (\alpha + u^*)(1 - u^*) \quad \beta > \gamma \quad \alpha < \frac{\beta-\gamma}{\gamma}$$

A time perturbation (t) is applied around E^* .

$$\begin{cases} \hat{u}(r, t) = u_0 e^{\lambda t} e^{ik\vec{r}} & \hat{v}(r, t) = v_0 e^{\lambda t} e^{ik\vec{r}} \\ |\hat{u}(r, t)| \ll u^* & |\hat{v}(r, t)| \ll v^* \end{cases} \quad (5)$$

where \vec{r} is space vector.

$$\begin{cases} g(u) = 1 - u & p(u) = \frac{u}{\alpha + u} \\ h(u) = \frac{ug(u)}{p(u)} = (1 - u)(\alpha + u) \\ f_u = p(u^*)h'(u^*) & f_v = -p(u^*) \\ g_u = \frac{\gamma^2}{\beta} & g_v = -\gamma & d = \frac{d_B}{d_A} & D = \begin{pmatrix} 1 & 0 \\ 0 & d \end{pmatrix} \end{cases} \quad (6)$$

The Jacobian matrix at the positive equilibrium is shown below:

$$J(E^*) = \begin{pmatrix} p(u^*)h'(u^*) & -p(u^*) \\ \frac{\gamma^2}{\beta} & -\gamma \end{pmatrix} \quad (7)$$

We can derive from the linear system:

$$|\lambda I - J + Dk^2| = 0 \quad (8)$$

The characteristic function of equation 5 is as follows:

$$\begin{cases} \lambda^2 + \lambda[k^2(1 + d) - (f_u + g_v)] + h(k^2) = 0 \\ h(k^2) = dk^4 - (df_u + g_v)k^2 + |J| \\ |J| = \gamma p(u^*) \left[\frac{\gamma}{\beta} - h'(E^*) \right] \end{cases} \quad (9)$$

The condition for a spatial mode defined by k to be unstable and thus to form a pattern in equation 9. $Re(\lambda)$ is a function of $h(k^2)$, and $h(k^2)$ is closely related with d . Therefore,

$d(d = \frac{d_B}{d_A})$ determines the range of $Re(\lambda)$ and the stability of the diffusion system. Next, we use numerical simulation²⁴ to verify whether the experimental system is unstable. First, we simplify equation 4 to the following form:

$$\begin{cases} f(u, v) = u(1 - u) - \frac{uv}{u + \alpha} \\ g(u, v) = \frac{\beta uv}{u + \alpha} - \gamma v \end{cases} \quad (10)$$

For the two-dimensional approximations, we use a uniform subdivision of the square by finite difference method.

$$\begin{cases} \Omega = [A, B] \times [A, B] & (x_i, y_j) = (ih + A, jh + A) \quad i, j = 0, \dots, J \\ h = \frac{B-A}{J} \end{cases} \quad (11)$$

$\vec{U}_{i,j}^n = (U_{i,j}^n, V_{i,j}^n)^T$ denotes the two-dimensional approximation at the point (x_i, y_j, t_n) . We also

carry out a uniform subdivision of the time interval $[0, T]$ with time levels $t_n = n\Delta t$, $n = 1, \dots, N$, so the time step is $\Delta t = \frac{T}{N}$. h is space step. Two-dimensional linear schemes are

revealed as the following general form. For $n = 1, \dots, N$ and $i, j = 0, \dots, J$ find $\{U_{i,j}^n, V_{i,j}^n\}$

$$\begin{cases} \partial_n U_{i,j}^n = \Delta h U_{i,j}^n + \hat{f}(U_{i,j}^n, U_{i,j}^{n-1}) \\ \partial_n V_{i,j}^n = d \Delta h V_{i,j}^n + \hat{g}(U_{i,j}^n, U_{i,j}^{n-1}) \end{cases} \quad (12)$$

$U_{i,j}^0 := u_0(x_i, y_j)$ and $V_{i,j}^0 := v_0(x_i, y_j)$ can be understood as the initial position of B and A entering the solution.

Function $[X, Y, U, V, reu, rev] = fd2d_predator_prey(\alpha, \beta, \gamma, d, a, b, h, T, \Delta t)$.

We simulated the experiment with fixed parameters. $\alpha = 0.4, \beta = 2.0, \gamma = 0.6, a = 0, b = 400, h = 1, \Delta t = \frac{1}{3}, d = \frac{1}{30}$ or $d = \frac{1}{20}$

$$\begin{cases} U_{i,j}^0 = \frac{6}{35} - 2 * 10^{-7} * (X(i,j) - 0.1 * Y(i,j) - 225) * \\ \quad (X(i,j) - 0.1 * Y(i,j) - 675) \\ V_{i,j}^0 = \frac{116}{245} - 3 * 10^{-5} * (X(i,j) - 450) - 1.2 * 10^{-4} * (Y(i,j) - 150) \end{cases} \quad (13)$$

$$\begin{cases} U_{i,j}^0 = \frac{6}{35} - 2 * 10^{-7} * (X(i,j) - 180)(X(i,j) - 520) - 6 * 10^{-7} * \\ \quad (Y(i,j) - 80)(X(i,j) - 200) \\ V_{i,j}^0 = \frac{116}{245} - 3 * 10^{-5} * (X(i,j) - 350) - 6 * 10^{-5} * (Y(i,j) - 235) \end{cases} \quad (14)$$

When $T = 1, T = 5, T = 10, \dots, T = 1000$, the simulated Turing pattern can be obtained.

Supplementary Fig. 19. Numerical simulation. a, Patterns exhibited by prey and predator of the system at $T = 1, 100, 600$ and 1000 , respectively. $d = \frac{1}{30}$, and the initial data is in line with the equation 13. **b,** Patterns exhibited by prey and predator of the system at $T = 1, 100, 600$ and 1000 , respectively. $d = \frac{1}{20}$, and the initial data is in line with the equation 14.”

“In order to further verify that the formation of Turing structure is determined by intrinsic property of the solution rather than an artifact that can appear due to the fabrication process, we used the drop coating and spin coating technique to prepare high-performance Turing structure films. The above conclusions are furtherly summarized from Supplementary Fig. 48. Once the difference of diffusion coefficient of related substances in homogeneous solution is modulated, Turing structure films can be obtained by either drop coating or spin coating process. Furthermore, the Turing structure films performances were found to be better than that of non-Turing structure films. Nevertheless, there is a gap between the properties of the films prepared by the above process and the films prepared by spray pyrolysis process. It is mainly due to the characteristics of spin coating and drop coating process, and we have not optimized the technology. For the drop coating process, although Turing structure film can also be prepared, the film area is usually small and the thickness is not easy to control. For the spin coating technique, the viscosity of the solution needs to be considered. If the process is optimized, a film with performance comparable to that of the spray pyrolysis film can be obtained. Accordingly, here we try to improve the spin

coating technique and give researchers more inspiration.

2. How the images for Turing structure were developed from the SEM images is not clear. For example, for Supplementary Fig. 4. The authors should include details in the revised manuscript.

Response: Thanks reviewer for the constructive suggestions. The development of Turing structure was mainly to adjust the diffusion coefficient difference between activator and inhibitor. Therefore, we have added more figures to illustrate. Furthermore, Supplementary Fig. 4 in our previous Supplementary Information was divided into Fig. 3e-g in the revised manuscript and Supplementary Fig. 16 in the revised Supplementary Information.

Fig. 3. Understanding the Turing structure by the reaction-diffusion equation. e, SEM image of Zn: Fe=1:2.4 film. f, the Turing interface film (Zn: Fe=1:3.5). g, the Turing interface film (Zn: Fe=1:4). In the fourth row of the figure, blue represents ZnFe_2O_4 , and pink represents $\alpha\text{-Fe}_2\text{O}_3$. Scale bars: c, d, e, 100 nm.

Supplementary Fig. 16. SEM images of Turing structure formation process. a, ZnFe_2O_4 film. b, the Turing interface film (Zn: Fe=1:3). c, $\alpha\text{-Fe}_2\text{O}_3$ film. In the fourth row of the figure, blue represents ZnFe_2O_4 and pink represents $\alpha\text{-Fe}_2\text{O}_3$. Scale bars: 100 nm.

3. In Supplementary Fig. 4, the presence of $\alpha\text{-Fe}_2\text{O}_3$ and ZnFe_2O_4 phases should be pointed clearly.

Response: Thanks for your constructive suggestions. As we adjusted the structure of the Supplementary Information, Supplementary Fig. 4 was divided into Fig. 3e-g (revised manuscript) and Supplementary Fig. 16 (revised Supplementary Information). In our experiment, as the concentration of $\text{Fe}[\text{C}_5\text{H}_7\text{O}_2]_3$ increase, $\alpha\text{-Fe}_2\text{O}_3$ phase becomes more stripe-like, and ZnFe_2O_4 phase exists as spinel octahedral nanoparticles. Therefore, the presence of $\alpha\text{-Fe}_2\text{O}_3$ and ZnFe_2O_4 phases has been clearly pointed out.

Fig. 3. Understanding the Turing structure by the reaction-diffusion equation. e, SEM image of Zn: Fe=1:2.4 film. f, the Turing interface film (Zn: Fe=1:3.5). g, the Turing interface film (Zn: Fe=1:4). In the fourth row of the figure, blue represents ZnFe_2O_4 , and pink represents $\alpha\text{-Fe}_2\text{O}_3$. Scale bars: c, d, e, 100 nm.

Supplementary Fig. 16. SEM images of Turing structure formation process. a, ZnFe_2O_4 film. b, the Turing interface film (Zn: Fe=1:3). c, $\alpha\text{-Fe}_2\text{O}_3$ film. In the fourth row of the figure, blue represents ZnFe_2O_4 and pink represents $\alpha\text{-Fe}_2\text{O}_3$. Scale bars: 100 nm.

4. The information about the LED light used in this manuscript is missing. What is the wavelength of LED used?

Response: We sincerely appreciate the reviewer for providing the constructive comments. Following the reviewer's suggestion, we have added the spectrum of the LED light in the revised supplementary information as shown in Supplementary Fig. 31a (Supplementary Fig. 29 in our previous Supplementary Information). And the following sentences were added in revised manuscript.

“LED illumination (LEDGUHON, China) was used to compare the improved performance under different light intensities.”

Supplementary Fig. 31. Photoelectrochemical activities under LED illumination. a, Spectral irradiance of a LED light.

5. How the authors calculated the ‘molecular electrostatic potential’ of the surfaces is not provided in Supplementary information. The authors should provide the details.

Response: We thank the reviewer for the good suggestion. The calculation description of the molecular electrostatic potential is added to the computational details of the revised manuscript.

“Gauss View can create a surface where the color is determined by the values of a second property, which implies that we can map the values of one property on an isosurface of a different property. The cubegen (a module of Gaussian software) was first employed to calculate the electron density isosurface (0.001 e/bohr^3). Subsequently, the electrostatic potential value of the point on the isosurface was calculated and displayed on the electron isosurface by color. Then, according to the electrostatic potential value of different molecules, the electron density surface was drawn.”

6. Supplementary Fig. 28f is not clear and difficult to understand.

Response: We appreciate the reviewer’s valuable comments that help to improve the quality of our work. We are quite sorry for the unclear display of this picture. We redraw it as shown in Supplementary Fig. 30f (Supplementary Fig. 28f in our previous Supplementary Information) to clearly show the charge separation efficiency and injection efficiency of the films. And the following sentences were added in revised manuscript.

“The $\alpha\text{-Fe}_2\text{O}_3$ film, the Turing interface film (Zn: Fe=1:3) and the Turing interface film (Zn: Fe=1:4) possessed a similar injection efficiency, which also revealed that the performance enhancement of Turing structure came from the improvement of separation efficiency.”

Supplementary Fig. 30. Photoelectrochemical activities. f, Charge separation efficiency and charge injection efficiencies of the photoelectrodes.

7. What is the area or the volume of the electrode materials used for the study of the amount of O_2

and H₂ produced should be included. For example, in Fig. Supplementary Fig. 30 b, c and d and other such plots in the whole manuscript in SI.

Response: Thanks for the reviewer's suggestion. We apologize that in our previous manuscript the related description was not explicit. We have revised the caption of Supplementary Fig. 30 (Supplementary Fig. 28 in our previous Supplementary Information), Supplementary Fig. 32 (Supplementary Fig. 30 in our previous Supplementary Information) and the description in the manuscript as follows:

“The Turing interface film (Zn: Fe=1:3) with an effective area of 1 cm² showed excellent stability (Supplementary Fig. 30c) at 1.23 V vs RHE, indicating their photochemical durability for water splitting.”

Supplementary Fig. 30. Photoelectrochemical activities. **a, b**, Current density-potential profiles of different films measured under LED illumination (296 mW cm^{-2}) and AM 1.5 G illumination (100 mW cm^{-2}), respectively. **c**, Stability test of the Turing interface film (Zn: Fe=1:3) with an effective area of 1 cm^2 for 8h under AM 1.5 G illumination (100 mW cm^{-2}) at 1.23 V vs. RHE. **d**, A photograph of water-splitting device. **e**, Gas evolution for PEC water splitting of the Turing interface film (Zn: Fe=1:3) with an effective area of 1 cm^2 were performed at 1.23 V RHE under AM 1.5 G illumination in 1M NaOH electrolyte. **f**, Charge separation efficiency and charge injection efficiencies of the photoelectrodes.

Supplementary Fig. 32. Faraday efficiency. a, Stability test of α-Fe₂O₃, ZnFe₂O₄ and Zn: Fe=1:4 film with an effective area of 1 cm² for 8h. b, c, d, Gas evolution curves under AM 1.5 G illumination (100 mW cm⁻²) at 1.23 V vs. RHE.

8. I do not understand the necessity to study the magnetic property of the material. Can the authors explain the same?

Response: We thank the reviewer for the valuable suggestion. We have deleted the magnetic property test and description of thin films.

Reviewer #3 (Remarks to the Author):

It is a nice piece of work on Interfacial engineering which would be useful for many energy applications. The work is interesting and can be published after incorporating these comments.

Response: We sincerely thank the reviewer's comments. The proposed suggestions are valuable and helpful for improving our work. We have carefully revised the manuscript and replied to the comments point-by-point shown below.

1. Though the role of Turing structures is known, authors should include a paragraph on the mechanism and role of such Turing structures in improving the charge recombination.

Response: We appreciate the reviewer's valuable comments very much. We are also sorry not to provide a clear description in the previous manuscript. In revised manuscript, we have made more profound and comprehensive discussions about the charge separation enhancement mechanism of the Turing structure film.

“Photoelectrochemical measurement and charge separation enhancement mechanism.

Having identified the essential roles of the interface structure, we propose an interface transport mechanism that shows a large difference in the separation efficiency of photogenerated carriers. In the photoelectrode film, electron-hole pairs are generated under light, followed by different charge transport and trapping processes under bias. In the Turing interface film (Fig. 6a), the generated electron-hole pairs separate effectively at the interface via an enhanced interface built-in electric field. In the conventional dual-phase interface film (Fig. 6b), a built-in electric field at the interface is weakened due to the existence of the incoherent interface or semi-coherent interface. Therefore, the electron and hole pairs cannot be effectively separated at the interface, which increases the probability of recombination within the particle. In the case of a non-Turing dual-phase interface film (Fig. 6c), the built-in electric field at the interface is also weakened, and the separated electrons and holes are recombined at the interface due to the mutual coating of the dual-phases. Above was further evaluated under AM 1.5 G irradiation with a standard three-electrode system in 1 M NaOH solution.”

Fig. 6. Enhancement mechanism of the Turing interface film. a, Schematic of the fundamental processes in the Turing interface film. Photon absorption and formation of the carriers and carrier separation at the interface. b, Schematic of the fundamental processes in the conventional dual-phase interface film. Photon absorption and formation of the carriers and carrier separation at the interface. c, Schematic of the fundamental processes in the non-Turing dual-phase interface

film. Photon absorption and formation of the carriers and carrier separation at the interface.

2. What is the role of FeCl_3 and $\text{Fe}(\text{NO}_3)_3$ during the film formation process.

Response: We appreciate the reviewer's valuable comments very much. In our work, the efficient separation of photogenerated carriers is achieved by constructing a Turing structure in the film. The bottleneck of realizing Turing structure is the regulation of the diffusion coefficient difference between activator and inhibitor in homogeneous solution, which has not been reported so far. Thus, taking the spinel film as a paradigm, we have studied in detail the relationship between the diffusion coefficients of the inhibitors and activators in the homogeneous solution, to further verify the formation of the Turing structure. Applying $\text{Fe}[\text{C}_5\text{H}_7\text{O}_2]_3$ as the Fe source, diffusion coefficient can be adjusted to an order of magnitude difference when the ratio of $\text{Fe}[\text{C}_5\text{H}_7\text{O}_2]_3$ and $\text{Zn}(\text{CH}_3\text{COO})_2$ is improved to reach a proper concentration. According to our experimental results as shown in Supplementary Fig. 15 (Supplementary Fig. 17 in our previous Supplementary Information) and Fig. 2f, the optimal concentration can make the diffusion coefficient of Fe more than thirty times that of Zn. As a control, in order to eliminate the influence of the difference in diffusion coefficient caused by the concentration gradient, we replaced $\text{Fe}[\text{C}_5\text{H}_7\text{O}_2]_3$ with FeCl_3 and $\text{Fe}(\text{NO}_3)_3$. Even if the concentration of Fe is increased, difference of diffusion coefficient fluctuates around 1

with no order of magnitude difference, and ultimately no Turing structure is formed. Therefore, we named the films prepared by FeCl_3 and $\text{Fe}(\text{NO}_3)_3$ as non-Turing dual-phase interface films and used them as comparison samples.

Supplementary Fig. 15. Ratio of Diffusion coefficient. M, $N = \ln(c_1 - c_2 / c_3 - c_4)$, c_1 and c_3 are the initial and final concentrations; c_2 and c_4 are the initial and final concentrations in the upper compartment).

Fig. 2. Hard-Soft-Acid-Base (HSAB) theory. f, Ratio of diffusion coefficients (Supplementary Fig. 18, for FeCl_3 , A is Zn: Fe=1:3 and B is Zn: Fe=1:4; for $\text{Fe}(\text{NO}_3)_3$, C is Zn: Fe=1:3 and D is Zn: Fe=1:4; for $\text{Fe}[\text{C}_5\text{H}_7\text{O}_2]_3$, E is Zn: Fe=1: 2, F is Zn: Fe=1:3, and G is Zn: Fe=1:4).

3. What is the reason of high output in the case of Turing interface film (Zn: Fe=1:3).

Response: We are greatly grateful to the reviewer's nice question and it is very useful for improving the quality of this work. Turing interface film (Zn: Fe=1:3) exhibits excellent photoelectrochemical water splitting and phenol oxidation due to its higher charge separation efficiency. More importantly, the unique interface features formed by the Turing structure. According to the analysis of abundant precise atomic interfaces in the film as shown in Fig. 5, semi-coherent interface with an interface strain close to zero is formed between the dual phases. Therefore, it is essential to consider the interface state in understanding the equilibrium energy band diagram as shown in Supplementary Fig. 38 (Supplementary Fig. 36 in our previous Supplementary Information). Thus, a strong interface built-in electric field is formed in the Turing interface film (Zn: Fe=1:3), which promotes the separation of photogenerated electron-hole pairs.

Fig. 5. Atom-level Turing interface. **a**, The oxygen-bridged coordination network. **b**, HAADF-STEM image of the Turing interface (Zn: Fe=1:3) and GPA analysis of different plane strains. The interface has been marked with a white dashed line in the strain graph, and the curve directly below graph corresponds to the strain value across the interface. **c**, TEM image of the conventional dual-phase interface (Zn: Fe=1:3) and GPA analysis of different plane strains. The interface has been marked with a purple dashed line in the strain graph, and the curve directly below graph corresponds to the strain value across the interface. **d**, Crystallographic models of the interface before phase transition and after phase transition. **e**, Raman spectra for the β - Fe_2O_3 powder, α - Fe_2O_3 powder and the unannealed film (β - Fe_2O_3). **f**, HAADF-STEM images for the unannealed and calcined films for the only iron source ($\text{Fe}[\text{C}_5\text{H}_7\text{O}_2]_3$). **g**, The image to a larger view in the dotted box (Supplementary Fig. 28a) and the semi-coherent interface. **h**, The image to a larger view in the dotted box (Supplementary Fig. 28b). **i**, GPA analysis of different plane strains for **g**. The interface has been marked with a white dashed line in the strain graph, and the curve directly below graph corresponds to the strain value across the interface. **j**, GPA analysis of different plane strains for **h**. The interface has been marked with a white dashed line in the strain graph, and the curve directly below graph corresponds to the strain value across the interface. **k**, the semi-coherent interface. Scale bars: **b**, 5 nm; **c**, 10 nm; **f**, 5 nm.

Supplementary Fig. 38. Interface states. **a**, Balanced energy band diagram of n-n heterojunction without interface states (coherent interface). **b**, Schematic diagram of the energy band of the n-n heterojunction when the interface state is considered (semi-coherent interface). **c**, Schematic diagram of the energy band of the n-n heterojunction when the interface states are considered (incoherent interface).

4. An explanation to be added to why charge transfer is non-existent in bulk Zn ferrite.

Response: It is a very good question, and we are sorry that the charge transfer process was not stated clearly in the previous manuscript. Generally speaking, transient photovoltage (TPV) is an effective method to study the dynamics of photogenerated carriers [r1, r2]. Photo-generated charge separation process includes drift and diffusion: the drift process refers to the fast separation process ($<10^{-5}$ s) that occurs within the particles; the main diffusion process is the charge transfer between particles over a long period of time ($>10^{-4}$ s). Thus, the photovoltage response includes two parts: rising and decay. The rising part of the photovoltage physically corresponds to the increase in the electron concentration of the conductive substrate of the electrode. Caused by the diffusion of photogenerated electrons to the substrate, the drop in photovoltage mainly corresponds to the recombination process of electrons leaving the conductive substrate. Therefore, charge transfer was tough in the bulk for ZnFe_2O_4 due to the fast recombination of photogenerated electrons and holes compared with the Turing interface film (Zn: Fe=1:3) (Fig. 6g). In the revised manuscript, the following text has been added:

Fig. 6. Enhancement mechanism of the Turing interface film. g. Transient photovoltage responses (TPV).

[r1] J.J. Yoo. et al. Efficient perovskite solar cells via improved carrier management. *Nature* 590, 587-593 (2021).

[r2] Q.Y. Wu. et al. A metal-free photocatalyst for highly efficient hydrogen peroxide photoproduction in real seawater. *Nat. Commun.* 12, 1-10 (2021).

“The dynamic properties of photoexcited charge separation and transfer processes were further explored by transient photovoltage technology (TPV) (Fig. 6g), which is widely used to understand the mechanism of semiconductor on the nanosecond time scale. Generally, the photo-generated charge separation process includes drift and diffusion: the drift process refers to the fast separation process ($<10^{-5}$ s) that occurs within the particles; the main diffusion process is the charge transfer between particles over a long period of time ($>10^{-4}$ s)³⁰. Thus, the photovoltage response includes two parts: rising and decaying. The rising part of the photovoltage physically corresponds to the increase in the electron concentration of the conductive substrate of the electrode. Caused by the diffusion of photogenerated electrons to the substrate, the drop in photovoltage mainly corresponds to the recombination process of electrons leaving the conductive substrate. Therefore, charge transfer was tough in the bulk for $ZnFe_2O_4$ due to the fast recombination of photogenerated electrons and holes compared with the Turing interface film (Zn: Fe=1:3).”

5. Can this be applied to perovskite solar cells as interfacial properties play a major role in the same.

Response: We thank the reviewer for the insightful suggestion. Following the comments, we performed additional experiments about perovskite solar cells. We use a traditional two-step solution method to prepare thin films [r1, r2]. The diffusion of substances is changed by introducing $Na[C_5H_7O_2]$ into the solution. Fig. Ra-c shows the SEM images of the prepared films without addition and with different concentrations of addition. There are obvious holes in the

traditional film without addition (Fig. Ra). After addition, the film presented a dense structure (Fig. Rb), and an obvious phase dispersion structure appears (Fig. Rc) when it continues to increase. Furthermore, XRD revealed that there was not much difference in the phases. Next, we assembled the device to evaluate photovoltaic parameters as shown in Fig. Re-g, our film exhibited champion PCE despite not being meticulously optimized. We are sure that this strategy may develop into a hot topic to study thin films.

Fig. R. Characterization and device performance CsPbBr₃ perovskite films. **a**, SEM images of CsPbBr₃ film. **b**, SEM images of CsPbBr₃ films prepared by incorporating a small amount of Na[C₅H₇O₂] in the precursor solution. **c**, SEM images of CsPbBr₃ film prepared by incorporating more Na[C₅H₇O₂] in the precursor solution. **d**, perovskite solar cells device architecture. **e**, XRD patterns. **f**, J-V curves. **g**, photovoltaic parameters under AM 1.5 G solar spectra with a light intensity of 100 mW cm⁻². J_{sc} (Short-circuit photocurrent), V_{oc} (Open-circuit photovoltage), FF (Fill Factor), PCE (Power Conversion Efficiency). Scale bars: 2.5 μm . A carbon back-electrode with an average area of 0.09 cm² was deposited on the CsPbBr₃ perovskite film by a doctor-blade coating method. The J-V plots were all recorded by the scans (a voltage step of 10 mV and a delay time of 50 ms) between -0.1 to 1.5 V and backward scan is applied for preliminary characterization, and overall efficiency is directly calculated from solar energy and discharge electrical energy. An Oriel 92251A-1000 sunlight simulator was used to provide the testing light AM 1.5G (100 mW cm⁻²). The light intensity was calibrated at 100 mW cm⁻² by the standard reference of a Newport 91150V silicon cell before use. (For review)

[r1] Q.X. Meng. et al. Simultaneous optimization of phase and morphology of CsPbBr₃ films via controllable ostwald ripening by ethylene glycol monomethylether/isopropanol Bi-solvent

engineering. Adv. Eng. Mater. 22, 2000162 (2020).

[r2] X.P. Han. et al. Carrier mobility enhancement in (121)-oriented CsPbBr₃ perovskite films induced by the microstructure tailoring of PbBr₂ precursor films. ACS Appl. Electron. Mater. 3, 373-384 (2021).

6. What is the surface roughness of the film and did it have any influence on the efficiency.

Response: We are greatly grateful to the reviewer for the nice question. We agree with the comment that the surface roughness of the film may influence the efficiency. In this regard, the surface roughness of the film will be analyzed in detail. In the previous manuscript, we measured the surface roughness of the Turing interface film (Zn: Fe=1:3 and Zn: Fe=1:4) by a step profiler as shown in revised Supplementary Fig. 17a and f (Supplementary Fig. 5i and j in our previous Supplementary Information), but we ignored the discussion. Therefore, we have added the relevant discussion through the atomic force microscope (AFM). Based on the surface roughness results of Supplementary Fig. 46 and 47, it can be concluded that no matter which iron source is used, the surface roughness is between 5-8 nm, and there is no distinct dependence between the surface roughness and the performance of the film. Therefore, we can confirm that the enhancement in the efficiency of the thin film is attributed to the formation of the bulk Turing interface. Accordingly, the figures have been added in revised Supplementary Information as Supplementary Fig. 46 and Supplementary Fig. 47. Meanwhile, the following text has been added in revised manuscript.

Supplementary Fig. 17. SEM images, elemental mapping and HAADF-STEM images of Turing structure film. **a**, Surface roughness of the Turing interface film (Zn: Fe=1:3) (Ra= 8.3 nm) measured by step profiler. **b**, **c**, **d**, **e**, Elemental mapping for the Turing interface film (Zn: Fe=1:3). **f**, Surface roughness of the Turing interface film (Zn: Fe=1:4) (Ra= 8.7 nm) measured by step profiler. **g**, **h**, **i**, **j**, Elemental mapping for the Turing interface film (Zn: Fe=1:4). **k**,

HAADF-STEM image of the Turing interface film (Zn: Fe=1:3). **l**, HAADF-STEM image of the Turing interface film (Zn: Fe=1:4). **m**, false-colored HAADF-STEM image. Purple represent α -Fe₂O₃ and green represent ZnFe₂O₄ by the crystal plane analysis in **l**. Scale bars: **b, c, d, e, g, h, i, j**, 250 nm; **k, l, m**, 20 nm.

Supplementary Fig. 46. AFM micrographs (2D and 3D) prepared from Fe[C₅H₇O₂]₃. a, ZnFe₂O₄ film. b, Zn: Fe=1:2.4 film. c, the Turing interface film (Zn: Fe=1:3). d, the Turing interface film (Zn: Fe=1:4). e, the Turing interface film (Zn: Fe=1:6). f, α -Fe₂O₃ film. g, average surface roughness (Ra).

Supplementary Fig. 47. AFM micrographs (2D and 3D) of non-Turing dual-phase interface films (Zn: Fe=1:3 and Zn: Fe=1:4) prepared from inorganic metal source. a, b, Fe(NO₃)₃. c, d, FeCl₃. e, average surface roughness (Ra).

“To exclude the influence of the surface structure of the film, we conducted a surface roughness measurement. According to the AFM micrographs (2D and 3D) in Supplementary Fig. 46 and 47, there is little difference between the Turing structure film and the non-Turing structure film, and the surface roughness is distributed at 5-8 nm. Therefore, we can be sure that the same preparation process has little effect on the surface roughness of the film. Moreover, there is no strong dependence between its performance and surface roughness. We confirm that the enhancement in the efficiency of the thin film is attributed to the formation of the bulk Turing interface.”

7. Do the fabrication of these films have a role in deciding the Turing structures.

Response: We are greatly grateful to the reviewer for the nice question and careful inspection. We have attempted to fabricate the films by different techniques and collect the SEM images which was found similar irrespective to the fabrication process. **Therefore, the Turing structure was an inherent feature of the homogeneous solution, which is not limited by the fabrication process.** The above conclusions are furtherly summarized from Supplementary Fig. 48. Once the difference of diffusion coefficient of related substances in homogeneous solution is modulated, Turing structure films can be obtained by either drop coating or spin coating process. Furthermore, the Turing structure films exhibited better performances than non-Turing structure films. Nevertheless, there is a gap between the properties of the films prepared by the above process and the films prepared

by spray pyrolysis process. It is mainly due to the characteristics of spin coating and drop coating process, and we have not optimized the technology. For the drop coating process, although Turing structure film can also be prepared, the film area is usually small and the thickness is not easy to control. For the spin coating technique, the viscosity of the solution needs to be considered. If the process is optimized, a film with performance comparable to that of the spray pyrolysis film can be obtained. Accordingly, here we try to improve the spin coating technique and give researchers more inspiration. By comparing the droplet evolution of mixed solutions composed of iron sources and $\text{Zn}(\text{CH}_3\text{COO})_2$ (Supplementary Fig. 49), we can draw the following conclusion: $\text{Fe}(\text{NO}_3)_3$ and FeCl_3 in four solvents show a diffusion coefficient difference close to 1, which does not appear regular spots or strips structure. $\text{Fe}[\text{C}_5\text{H}_7\text{O}_2]_3$ is used as a source of iron, and ethylene glycol or acetylacetone as a solvent, which will not form a Turing structure. While it exhibits a regular structure in DMF solution. Simultaneously, introducing small amount of surfactant into the methanol solution does not affect the diffusion coefficient of the ions, so as to adjust the viscosity of the solution, it will be easier to spin coating into a high-quality film. This further confirms that the Turing structures of film is the intrinsic property of the solution. And the following sentences were added in revised supplementary information.

Supplementary Fig. 48. Different fabrication strategies were used to verify the homogeneous solution characteristics of Turing structure and non-Turing structure. Drop coating: a, The shape evolution of droplets (Zn: Fe=1:3, $\text{Fe}[\text{C}_5\text{H}_7\text{O}_2]_3$). **b,** SEM image and elemental mapping of Turing structure film for above solution. **c,** The shape evolution of droplets (Zn: Fe=1:3, $\text{Fe}(\text{NO}_3)_3$). **d,** SEM image and elemental mapping of non-Turing dual-phase interface film for above solution. **e,** The shape evolution of droplets (Zn: Fe=1:3, FeCl_3). **f,** SEM image and elemental mapping of non-Turing dual-phase interface films for above solution. **g,** XRD patterns for Turing structure film and non-Turing structure film. **h,** Current density-potential profiles of Turing structure film and non-Turing structure film. **Spin coating: i,** The shape evolution of

droplets (Zn: Fe=1:3, $\text{Fe}[\text{C}_5\text{H}_7\text{O}_2]_3$). **j**, SEM image and elemental mapping of Turing structure film for above solution. **k**, The shape evolution of droplets (Zn: Fe=1:3, $\text{Fe}(\text{NO}_3)_3$). **l**, SEM image and elemental mapping of non-Turing dual-phase interface films for above solution. **m**, The shape evolution of droplets (Zn: Fe=1:3, FeCl_3). **n**, SEM image and elemental mapping of non-Turing dual-phase interface films for above solution. **g**, XRD patterns for Turing structure film and non-Turing structure film. **h**, Current density-potential profiles of Turing structure film and non-Turing structure film. Scale bars: **b, d, f, j, l, n**, 200 nm.

Supplementary Fig. 49. The shape evolution of droplets with different solvents.

“In order to further verify that the formation of Turing structure is determined by intrinsic property of the solution rather than an artifact that can appear due to the fabrication process, we used the drop coating and spin coating technique to prepare high-performance Turing structure films. The above conclusions are furtherly summarized from Supplementary Fig. 48. Once the difference of diffusion coefficient of related substances in homogeneous solution is modulated, Turing structure films can be obtained by either drop coating or spin coating process. Furthermore, the Turing structure films performances were found to be better than that of non-Turing structure films. Nevertheless, there is a gap between the properties of the films prepared by the above process and the films prepared by spray pyrolysis process. It is mainly due to the characteristics of spin coating and drop coating process, and we have not optimized the technology. For the drop coating process, although Turing structure film can also be prepared, the film area is usually small and the thickness is not easy to control. For the spin coating technique, the viscosity of the solution needs to be considered. If the process is optimized, a film with performance comparable to that of the spray pyrolysis film can be obtained. Accordingly, here we try to improve the spin

coating technique and give researchers more inspiration.

By comparing the droplet evolution of mixed solutions composed of iron sources and $\text{Zn}(\text{CH}_3\text{COO})_2$ (Supplementary Fig. 49), we can draw the following conclusion: $\text{Fe}(\text{NO}_3)_3$ and FeCl_3 in four solvents show a diffusion coefficient difference close to 1, which does not appear regular spots or strips structure. $\text{Fe}[\text{C}_5\text{H}_7\text{O}_2]_3$ is used as a source of iron, and ethylene glycol or acetylacetone as a solvent, which will not form a Turing structure. While it exhibits a regular structure in DMF solution. Simultaneously, introducing small amount of surfactant into the methanol solution does not affect the diffusion coefficient of the ions, so as to adjust the viscosity of the solution, it will be easier to spin coating into a high-quality film. This further confirms that the Turing structures of film is the intrinsic property of the solution.”

8. What is the thickness of the films.

Response: We are greatly grateful to the reviewer for the nice question. We optimized the thickness of the film to the best in the experiment, the thickness of the film was maintained at 300 nm as shown in Supplementary Fig. 18 (Supplementary Fig. 5 in our previous Supplementary Information), but we are sorry that we have less analysis of the film thickness. Here, we study that inorganic Turing films have high photogenerated charge separation efficiency. In order to highlight the advantages of this kind of films, we prepared non-Turing dual-phase interface film and conventional dual-phase interface film. Furtherly, we add the thickness of conventional dual-phase interface film in Supplementary Fig. 5 (Supplementary Fig. 7 in our previous Supplementary Information) and non-Turing dual-phase interface film in Supplementary Fig. 6 (Supplementary Fig. 8 in our previous Supplementary Information) in the revised Supplementary information. Considering light absorption, the thickness of the film needs to be kept moderate, so we show the summarized relationship between performance and thickness in Supplementary Fig. 35b (Supplementary Fig. 33 in our previous Supplementary Information).

Supplementary Fig. 18. Side-view SEM images and elemental mapping. a, $\alpha\text{-Fe}_2\text{O}_3$ film. **b,** ZnFe_2O_4 film. **c,** the Turing interface film (Zn: Fe=1:3). **d,** the Turing interface film (Zn: Fe=1:4). Scale bars: **a, b, c, d,** 300 nm.

Supplementary Fig. 5. Side-view SEM images, elemental mapping and TEM images of conventional dual-phase interface film. **a**, Side-view SEM images and elemental mapping of Zn: Fe=1:3 film. **b**, Side-view SEM images and elemental mapping of Zn: Fe=1:4 film. **c**, Control the film thickness by changing the deposition time. **d**, TEM image and HR-TEM image of Zn: Fe=1:3 film (Scrape it off from the conventional dual-phase interface film by electrophoretic deposition, $n_{\text{ZnFe}_2\text{O}_4} : n_{\alpha\text{-Fe}_2\text{O}_3} = 2:1$). **e**, TEM image and HR-TEM image of Zn: Fe=1:4 film (Scrape it off from the conventional dual-phase interface film by electrophoretic deposition, $n_{\text{ZnFe}_2\text{O}_4} : n_{\alpha\text{-Fe}_2\text{O}_3} = 1:1$). ZnFe_2O_4 is yellow, while $\alpha\text{-Fe}_2\text{O}_3$ is pink. Scale bars: **a, b, c**, 5 μm ; **d, e**, 20 nm.

Supplementary Fig. 6. SEM images and XRD patterns of non-Turing dual-phase interface

films (Zn: Fe=1:3 and Zn: Fe=1:4) prepared from inorganic metal source. a, b, $\text{Fe}(\text{NO}_3)_3$. c, d, FeCl_3 . e, Side-view SEM images and elemental mapping of Zn: Fe=1:3 film prepared by $\text{Fe}(\text{NO}_3)_3$. f, Side-view SEM images and elemental mapping of Zn: Fe=1:3 film prepared by FeCl_3 . g, XRD patterns for a, b, c and d. f, the photocurrent density (1.6 V vs. RHE, AM 1.5G, 100 mW cm^{-2}) for a, b, c and d. Scale bars: a, b, c, d, 100 nm; e, f, 500 nm.

Supplementary Fig. 35. b, Curves of relationship between photocurrents at 1.6 V vs. RHE and film thickness.

“In addition, the optimized film thicknesses were kept at 300 nm, 300 nm and 3.2 μm for the Turing interface films, non-Turing dual-phase interface films and conventional dual-phase interface films (Supplementary Fig. 35b).”

9. Similar reports should be compared in the discussion.

Response: We are greatly grateful to the reviewer for the nice question and it is useful for improving the quality of this work. Our current work is completely different from the previous work in mechanism: the previous work focused on the interfacial reaction in heterogeneous solution [r1, r2]. However, to date, there is a huge lacuna in designing and developing an inorganic Turing structure (spots or stripes) in homogeneous solutions, which is a huge challenge because of the similar diffusion coefficients for most small molecule weight species. Thus, our present work uses the re-coordination thought based on hard-soft acid-base theory to adjust the diffusion coefficient difference of the substance in a homogeneous solution. In view of the universality of this strategy, we have verified it in multiple film systems, which provides a new idea for the development of high-performance films in the future. The following sentences were added in revised manuscript.

[r1] Z. Tan, S.F. Chen, X.S. Peng, L. Zhang, C.J. Gao. Polyamide membranes with nanoscale Turing structures for water purification. *Science* **360**, 518-521 (2018).

[r2] X.L. Zhang. et al. An efficient Turing-type $\text{Ag}_2\text{Se-CoSe}_2$ multi-interfacial oxygen-evolving electrocatalyst. *Angew. Chem. Int. Ed.* **60**, 6553-6560 (2021).

“Tan²² used a facile route based on aqueous-organic interfacial polymerization to generate Turing-type polyamide membranes for water purification, and these membranes exhibit excellent water-salt separation performance. Zhang²⁴ reported a cation exchange approach in the heterogeneous solvent of diethylenetriamine and deionized water to produce Turing-type Ag₂Se on CoSe₂ nanobelts relied on diffusion-driven instability, which is highly effective in catalyzing the oxygen evolution reaction (OER) in alkaline electrolytes with an 84.5% anodic energy efficiency. It should be noted that the above case studies the related potential of the Turing structure in heterogeneous solution.”

REVIEWER COMMENTS

Reviewer #1 (Remarks to the Author):

The authors replied to the referees' comments point-by-point and gave appropriate answers. In the revised manuscript and the reply, the authors have largely improved the theoretical support on the evidence of the Turing pattern formation with the numerical simulation based on the predator-prey model. The difference in the diffusion constant between Fe and Zn can be a reasonable condition of Turing pattern formation. They obtained numerical results that bear some resemblance to the observed patterns. Although it is not strong evidence of the Turing pattern, it may include the truth. We cannot judge the correctness at this state, but I think it is constructive for the related fields to propose one possibility and stimulate further investigations on this topic. In this sense, I think the manuscript can be considered for publication in Nature Communications.

However, it is desirable to add reasonable explanations on the following points.

(1) There are various types of models that generate the Turing patterns. Why is the current predator-prey model the most plausible model?

(2) There are various parameters in the present model. What are their relationships to the experiment?

Minor point:

I do not understand the following sentence or equation

Function $[X, Y, U, V, reu, rev] = fd2d_predator_prey(\alpha, \beta, \gamma, d, a, b, h, T, \Delta t)$,

just below Eq. (12).

Please correct it appropriately.

Reviewer #2 (Remarks to the Author):

After carefully going through the manuscript, I believe that the manuscript can be accepted for publication.

Reviewer #3 (Remarks to the Author):

All the reviews have been addressed. The revised manuscript can be accepted now.

REVIEWER COMMENTS

Reviewer #1 (Remarks to the Author):

The authors replied to the referees' comments point-by-point and gave appropriate answers. In the revised manuscript and the reply, the authors have largely improved the theoretical support on the evidence of the Turing pattern formation with the numerical simulation based on the predator-prey model. The difference in the diffusion constant between Fe and Zn can be a reasonable condition of Turing pattern formation. They obtained numerical results that bear some resemblance to the observed patterns. Although it is not strong evidence of the Turing pattern, it may include the truth. We cannot judge the correctness at this state, but I think it is constructive for the related fields to propose one possibility and stimulate further investigations on this topic. In this sense, I think the manuscript can be considered for publication in Nature Communications.

However, it is desirable to add reasonable explanations on the following points.

- (1) There are various types of models that generate the Turing patterns. Why is the current predator-prey model the most plausible model?
- (2) There are various parameters in the present model. What are their relationships to the experiment?

Minor point:

I do not understand the following sentence or equation

Function $[X, Y, U, V, reu, rev] = fd2d_predator_prey(\alpha, \beta, \gamma, d, a, b, h, T, \Delta t)$, just below Eq. (12).

Please correct it appropriately.

Reviewer #2 (Remarks to the Author):

After carefully going through the manuscript, I believe that the manuscript can be accepted for publication.

Reviewer #3 (Remarks to the Author):

All the reviews have been addressed. The revised manuscript can be accepted now.

Point-by-point response for *Nature Communications* manuscript

(ID: NCOMMS-21- 39616A)

Manuscript Type: Article

Title: Homogeneous solution assembled Turing structures with near zero strain semi-coherence interface.

Author(s): Yuanming Zhang, Ningsi Zhang, Yong Liu, Yong Chen, Huiting Huang, Wenjing Wang, Xiaoming Xu, Yang Li, Fengtao Fan, Jinhua Ye, Zhaosheng Li, Zhigang Zou

General response: We sincerely thank the editor, editorial staff and all reviewers for their critical comments that we have based on to improve the quality of our manuscript. The manuscript has been modified point-by-point after addressing all the suggestions as listed below.

(Our response is given in blue, some key sentences are highlighted in yellow and the corrections in the revised manuscript are shown in red)

Reviewer #1 (Remarks to the Author):

The authors replied to the referees' comments point-by-point and gave appropriate answers. In the revised manuscript and the reply, the authors have largely improved the theoretical support on the evidence of the Turing pattern formation with the numerical simulation based on the predator-prey model. The difference in the diffusion constant between Fe and Zn can be a reasonable condition of Turing pattern formation. They obtained numerical results that bear some resemblance to the observed patterns. Although it is not strong evidence of the Turing pattern, it may include the truth. We cannot judge the correctness at this state, but I think it is constructive for the related fields to propose one possibility and stimulate further investigations on this topic. In this sense, I think the manuscript can be considered for publication in *Nature Communications*.

Response: We are very grateful for the reviewer. The reviewer's comments are very constructive and insightful to further improve the quality of the manuscript. We have made a point-by-point responses to address the reviewer's concerns.

In a reaction-diffusion system, the stable state can destabilize under certain conditions and spontaneously create space stable pattern. Therefore, the generation of the Turing pattern corresponds to the coupling of a nonlinear reaction kinetic process and a diffusion process. From the understanding of the mathematical mechanism, the stable constant equilibrium state of the ordinary differential systems undergoes a stability reversal after the addition of diffusion. Therefore, we also believe that in the future, more systems will be able to design Turing structure from the basic mathematical point of view, so as to promote the development of this field.

However, it is desirable to add reasonable explanations on the following points.

(1) There are various types of models that generate the Turing patterns. Why is the current predator-prey model the most plausible model?

Response: We thank for the reviewer's meaningful comments.

As research progresses, various types of models that can generate Turing patterns are developed, including Lotka-Volterra (predator-prey), Gierer-Meinhardt, Lengyel-Epstein, Thomas and so on [r1]. In the early years, Turing's reaction-diffusion model for pattern formation was attracted more attention from theoretical biology and applied mathematics [r2]. Alan Turing expressed this mechanism of pattern formation in terms of simultaneous differential equations of the form:

$$\begin{cases} \frac{\partial u}{\partial t} = f(u, v) + d_u \Delta u \\ \frac{\partial v}{\partial t} = g(u, v) + d_v \Delta v \end{cases}$$

where u and v is the vector of concentration, the functions $f(u, v)$ and $g(u, v)$ represent the reaction kinetics (diffusion term), d_u and d_v are the diffusion coefficients of u and v , respectively.

Therefore, no matter what kind of model, the most basic and primitive mathematical formula is the above equation. And the necessary conditions for the formation of Turing patterns are the following:

- (1) auto-catalysis and cross-catalysis must exist between an activator and an inhibitor;
- (2) the diffusion of the inhibitor must be much faster than that of the activator [r3-r5].

In our experiment, substance A (Fe) and substance B (Zn) with large difference in diffusion coefficient correspond to inhibitor and activator, respectively. The two substances are simultaneously affected by cross-diffusion according to the hard-soft acid-base theory. At the same time, the reacted substance concentrations are all positive values. These features fit best with classical predator-prey models. The prey acts as an activator, seeking to reproduce and increase their numbers, while the predator acts as an inhibitor, keeping populations in check. More importantly, when conducting relevant research in the fields of economy, physics and chemistry, scholars often select predator-prey model and convert the differential equation into difference equation for numerical simulation [r6, r7]. The difference equation can well continue some properties of the differential equation, such as instability. Although other models can also be numerically simulated, they are difficult to match the complex conditions of chemical reactions.

[r1] F.Q. Yi, J.J. Wei, H.P. Shi. Global asymptotical behavior of the Lengyel-Epstein reaction-diffusion system. *Appl. Math. Lett.* **22**, 52-55 (2009).

[r2] Z. Tan, S.F. Chen, X.S. Peng, L. Zhang, C.J. Gao. Polyamide membranes with nanoscale Turing structures for water purification. *Science* **360**, 518-521 (2018).

[r3] A.J. Koch, H. Meinhardt. Biological pattern formation: from basic mechanisms to complex structures. *Rev. Mod. Phys.* **66**, 1481-1507 (1994).

- [r4] S. Kondo, T. Miura. Reaction-diffusion model as a framework for understanding biological pattern formation. *Science* **329**, 1616-1620 (2010).
- [r5] Y. Fuseya, H. Katsuno, K. Behnia, A. Kapitulnik. Nanoscale Turing patterns in a bismuth monolayer. *Nat. Phys.* **17**, 1031-1036 (2021).
- [r6] M. R. Garvie. Finite-difference schemes for reaction-diffusion equations modeling predator-prey interactions in MATLAB. *Bull. Math. Biol.* **69**, 931-956 (2007).
- [r7] G.P. Hu, Z.S. Feng. Turing instability and pattern formation in a strongly coupled diffusive predator-prey system. *Int. J. Bifurcation and Chaos.* **30**, 2030020-1-15 (2020).

The related discussion has been added in the revised manuscript.

“In our experiment, substance A (Fe) and substance B (Zn) with large difference in diffusion coefficient correspond to inhibitor and activator, respectively. The two substances are simultaneously affected by cross-diffusion according to the hard-soft acid-base theory. At the same time, the reacted substance concentrations are all positive values. These features fit best with classical predator-prey models. More importantly, when conducting relevant research in the fields of economy, physics and chemistry, scholars often select predator-prey model and convert the differential equation into difference equation for numerical simulation^{27, 28}. The difference equation can well continue some properties of the differential equation, such as instability. Although other models can also be numerically simulated, they are difficult to match the complex conditions of chemical reactions.”

(2) There are various parameters in the present model. What are their relationships to the experiment?

Response: Thanks for the reviewer’s valuable comments.

In the derivation part of the predator-prey model in the Supplementary Information, we verified that the occurrence of Turing instability is closely related to the difference of the diffusion coefficients of the two substances, so the difference of the diffusion coefficients of the substances d is a key parameter connecting the experiment and the model ($d = \frac{d_B}{d_A}$) according to the according to the following predator-prey model:

$$\begin{cases} u_t = d_A \Delta u + u(1-u) - \frac{uv}{u+\alpha} & x \in \Omega, t > 0 \\ v_t = d_B \Delta v + \frac{\beta uv}{u+\alpha} - \gamma v & x \in \Omega, t > 0 \\ \frac{\partial u}{\partial n} = \frac{\partial v}{\partial n} = 0 & x \in \partial\Omega, t > 0 \\ u(x, 0) = u_0 \geq 0 \quad v(x, 0) = v_0 \geq 0 & x \in \Omega \end{cases}$$

where u and v represent the diffusion functions of B and A in the experimental system respectively, d_A and d_B are the diffusion coefficients of A and B, respectively. Ω is a fixed bounded domain (FTO substrate boundary), n is the outward unit normal vector of the boundary, Δ is Laplace operator, $\frac{\partial u}{\partial n} = \frac{\partial v}{\partial n} = 0$ is the homogeneous boundary condition, $u(x, 0) = u_0 \geq 0$

and $v(x, 0) = v_0 \geq 0$ are continuous functions.

In addition, $U_{i,j}^0$ and $V_{i,j}^0$ can be understood as the initial position of B and A entering the solution. Further, the meaning of each parameter in MATLAB is also listed. **The related discussion has been added in the revised Supplementary Information (Numerical Simulation section).**

“where u and v represent the diffusion functions of B and A in the experimental system respectively, d_A and d_B are the diffusion coefficients of A and B, respectively. Ω is a fixed bounded domain (FTO substrate boundary), n is the outward unit normal vector of the boundary, Δ is Laplace operator, $\frac{\partial u}{\partial n} = \frac{\partial v}{\partial n} = 0$ is the homogeneous boundary condition, $u(x, 0) = u_0 \geq 0$ and $v(x, 0) = v_0 \geq 0$ are continuous functions.”

“ $U_{i,j}^0 := u_0(x_i, y_j)$ and $V_{i,j}^0 := v_0(x_i, y_j)$ can be understood as the initial position of B and A entering the solution.”

“where X and Y are plane coordinates in simulation space (solution), U and V are spatiotemporal response of two diffusing substances, reu and rev represent the result of each iteration, α , β and γ represent the parameter in formula 10, d represent the ratio of the diffusion coefficients of two substances, a and b represent spatial positive domain (solution), h is space step, T is the maximum time for system simulation and Δt is time step.”

Minor point:

I do not understand the following sentence or equation

Function $[X, Y, U, V, reu, rev] = fd2d_predator_prey(\alpha, \beta, \gamma, d, a, b, h, T, \Delta t)$, just below Eq. (12).

Please correct it appropriately.

Response: Thanks for the reviewer’s valuable comments. We are sorry for the unclear description of this sentence. It is the MATLAB code in which we perform numerical simulations. Therefore, we added a more detailed description in the revised Supplementary Information (Numerical Simulation section).

“Description in the MATLAB:

Function $[X, Y, U, V, reu, rev] = fd2d_predator_prey(\alpha, \beta, \gamma, d, a, b, h, T, \Delta t)$.

where X and Y are plane coordinates in simulation space (solution), U and V are spatiotemporal response of two diffusing substances, reu and rev represent the result of each iteration, α , β and γ represent the parameter in formula 10, d represent the ratio of the diffusion coefficients of two substances, a and b represent spatial positive domain (solution), h is space step, T is the maximum time for system simulation and Δt is time step.”

Reviewer #2 (Remarks to the Author):

After carefully going through the manuscript, I believe that the manuscript can be accepted for publication.

Response: We highly appreciate the reviewer's recommendation very much.

Reviewer #3 (Remarks to the Author):

All the reviews have been addressed. The revised manuscript can be accepted now.

Response: We sincerely thank the reviewer for the consideration for the publication of our research work.

REVIEWERS' COMMENTS

Reviewer #1 (Remarks to the Author):

The authors have revised the manuscript appropriately according to the reviewer's comments. I think now the revised manuscript can be accepted for publication in Nature Communications.